# QoQ-Med: Building Multimodal Clinical Foundation Models with Domain-Aware GRPO Training

**Wei Dai, Peilin Chen, Chanakya Ekbote, Paul Pu Liang**
MIT Media Lab and MIT EECS
{dvdai, peili, cekbote, ppliang}@mit.edu

## Abstract

Clinical decision-making routinely demands reasoning over heterogeneous data, yet existing multimodal language models (MLLMs) remain largely vision-centric and fail to generalize across clinical specialties. To bridge this gap, we introduce **QoQ-Med-7B/32B**, the first open generalist clinical foundation model that jointly reasons across medical images, time-series signals, and text reports. QoQ-Med is trained with **D**omain-aware **R**elative **P**olicy **O**ptimization (DRPO), a novel reinforcement-learning objective that hierarchically scales normalized rewards according to domain rarity and modality difficulty, mitigating performance imbalance caused by skewed clinical data distributions. Trained on 2.61 million instruction tuning pairs spanning 9 clinical domains, we show that DRPO training boosts diagnostic performance by 43% in macro-F1 on average across all visual domains as compared to other critic-free training methods like GRPO. Furthermore, with QoQ-Med trained on intensive segmentation data, it is able to highlight salient regions related to the diagnosis, with an IoU 10x higher than open models while reaching the performance of OpenAI o4-mini. To foster reproducibility and down-stream research, we release (i) the full model weights, (ii) the modular training pipeline, and (iii) all intermediate reasoning traces at this link.

## 1 Introduction

Clinical diagnosis has evolved significantly over the past decade, with numerous computational models developed to assist clinicians in organizing patient records [87, 5], formulating diagnoses [19, 42], interpreting clinical images [26, 83], and other clinical tasks [12]. These advancements have substantially improved healthcare efficiency and accuracy across multiple specialties [21, 77]. Recently, the emergence of powerful generalist reasoning models such as OpenAI o3 [61] and Deepseek R1 [29] have inspired efforts to create specialized clinical reasoning systems [46, 91, 65] capable of answering complex clinical questions and generating comprehensive clinical reports [20, 90]. Reasoning allows models to think explicitly in a more logical and systematic way with evidence from the inputs and their own knowledge [34], all of which are essential for clinical diagnosis [55, 49].

However, building effective models to support clinical diagnosis presents several significant challenges. First, clinical data spans multiple modalities across 1D (ECG, EEG), 2D (Chest X-ray, dermoscopy, mammography), and 3D (CT Scans, MRI). Models like BiomedGPT [90] and Med-Flamingo [56] have integrated 2D and 3D data within one vision encoder, but no existing model has been able to integrate both 1D sensor data with 2D/3D images. The heterogeneity across specialties and modalities [28, 17] often leads to settings where modalities compete rather than synergize, leading to suboptimal performance [1, 36, 52, 89]. This necessitates careful retraining or fine-tuning strategies to balance heterogeneous distributions while enriching these models with clinical knowledge.

Secondly, conventional training methodologies typically constrain models to generate single, definitive answers without revealing their underlying analytical process [80, 76, 45]. This "black box" approach significantly impedes the practical adoption of AI systems in clinical settings, as healthcare professionals might hesitate to trust diagnostic suggestions without understanding the reasoning that produced them [72]. Transparency in the decision-making process is not merely a preference but a

39th Conference on Neural Information Processing Systems (NeurIPS 2025).

necessary component for responsible clinical implementation, regulatory compliance, and effective human-AI collaboration in healthcare environments [6, 66, 11].

In this work, we introduce QoQ-Med: a generalist clinical multimodal foundation model with precise reasoning capabilities spanning clinical images, time series data, and textual records across 9 clinical domains. Our work makes two primary contributions:

1. Firstly, to tackle the challenges associated with balancing heterogeneous data for balanced and efficient training across 1D to 3D data, we propose **Domain-aware Group Relative Policy Optimization (DRPO)**. DRPO employs hierarchical scaling based on the domain of the input data, which encourages the model's learning on scarce and hard domains, allowing balanced learning across difficulty levels. Our empirical evaluation demonstrates that DRPO consistently outperforms established RL approaches in diverse multi-domain settings, with up to 43% improvement in average F1 score across 8 clinical vision modalities.

2. To tackle the second challenge of expert interpretability, we design and release one of the first multimodal clinical reasoning models, namely **QoQ-Med-7B/32B** (**Q**wen **O**mni-Reasoning on **Med**ical **Q**uestions), that integrates visual, time series, and textual data for comprehensive analysis of clinical records, facilitating more holistic diagnostic reasoning. QoQ-Med is trained to highlight salient regions in the visual input data, advancing the interpretability while allowing the clinician to check the model's diagnosis with ease. To the best of our knowledge, QoQ-Med is currently the largest open-source multimodal reasoning model for clinical diagnosis, and the only MLLM that integrates time series data (ECG) with traditional clinical vision modalities.

Finally, we publicly release our model, training pipeline, and reasoning traces generated by the model across 2.61 million question-answer pairs at this link. This marks one of the largest resources for transparent and reproducible multimodal reasoning in the clinical domain.

## 2 Related Work

### 2.1 Multimodal Large Language Models (MLLMs) for clinical diagnosis

Recent work has adapted vision–language interfaces to the medical domain, yielding models such as LLaVa-Med [48], RadLM [90], and Med-Flamingo [56]. These models couple frozen LLM backbones with image encoders and are trained on radiology or pathology visual-question-answering and report-generation benchmarks [24, 39, 92, 88, 86]. Although these systems demonstrate impressive zero-shot understanding, their training corpora are dominated by single-institution chest X-rays, retinal photographs, and pathology slides, resulting in limited generalization to demographic diversity and poor robustness to real-world distribution [44, 63, 51]. GEM [47] is the only MLLM incorporating ECG data, but the training focus is purely ECG, which does not provide a comprehensive diagnosis aggregating multiple sources. Our work addresses these gaps by assembling a richer corpus spanning imaging, time-series, and text, and by designing an architecture that natively models medical time-series alongside traditional modalities.

### 2.2 LLM reasoning with reinforcement learning

The introduction of instruction tuning precipitated a rapid shift from supervised fine-tuning to reinforcement learning pipelines. Proximal Policy Optimization (PPO) [74] as popularized by InstructGPT, trains LLMs against a reward model under a KL penalty to a frozen reference, with an auxiliary critic estimating advantages [62]. While effective, PPO's critic incurs substantial memory and computation costs and can destabilise multi-task optimization [73]. To reduce overhead, critic-free objectives such as Direct Preference Optimization (DPO) [70] and Group Relative Policy Optimization (GRPO) [75] have emerged, matching PPO's alignment quality with a simple classification loss. GRPO, in particular, has been widely used in the training of recent SoTA models, such as DeepSeek R1 [29] and Qwen-3 [79]. However, removing the critic also eliminates per-sample re-weighting, causing it to overfit on easy, abundant samples [33]. Classic deep-RL work explored adaptive rescaling through task-wise normalization in IMPALA [27] and the PopArt [31]. However, these techniques have not been adapted to LLMs or extended to capture fine-grained intra-domain differences. We reinstate that flexibility by learning both inter-domain and intra-domain scaling factors within a critic-free RLHF pipeline, combining the efficiency advantages of GRPO with the adaptive weighting capabilities offered by critic-based methods.

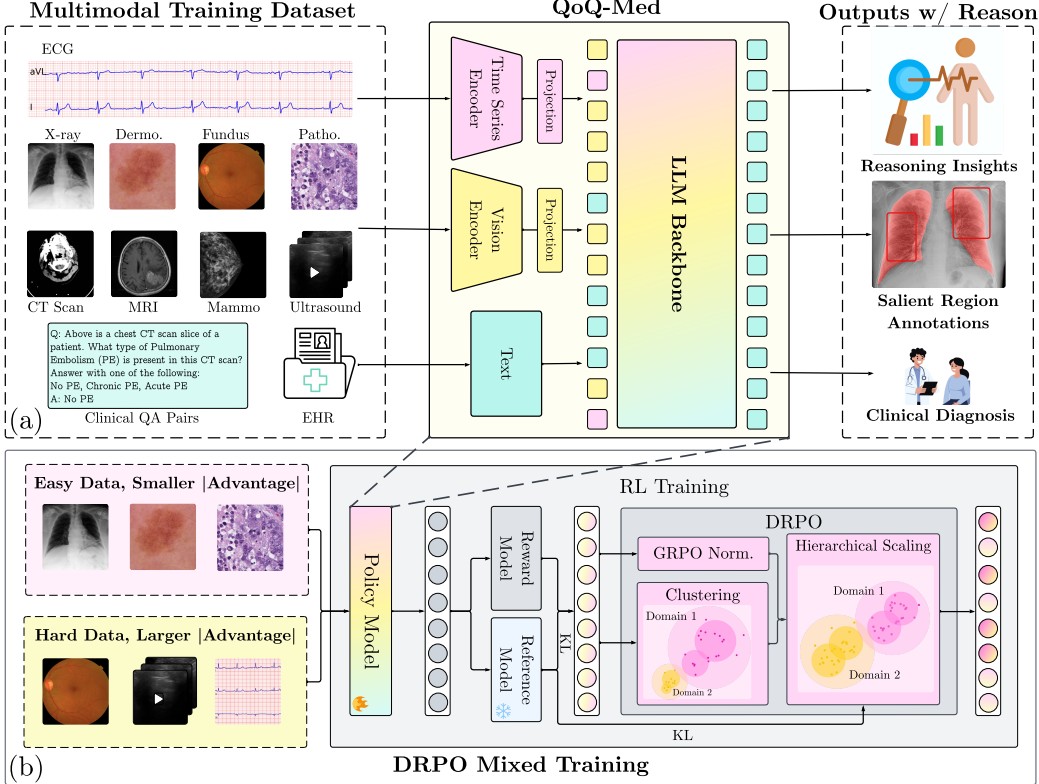

Figure 1: **(a) Overview of QoQ-Med.** The training corpus spans 11 clinical domains, including structured waveforms (e.g., ECG), diverse imaging modalities, electronic health records, and curated clinical QA pairs. Modality-specific encoders convert inputs into token embeddings that are linearly projected into a common space and interleaved with text tokens before entering the LLM backbone. The model then autoregressively produces (i) an explainable chain-of-thought, (ii) bounding-box annotations highlighting salient regions, and (iii) a concise clinical diagnosis. **(b) Overview of DRPO Training.** DRPO builds on top of the critic-free RL training method GRPO. The model's answer is first rated by a reward model before going through standard normalization. Then, a clustering-based scaling is performed on top of domain-wise scaling, both of which encourage the model to focus on scarce, hard examples across domains.

## 3  Method

In this section, we first define our problem as a multimodal diagnosis question answering task, before describing how we integrated time series alongside vision inputs into a single unified model. Finally, we demonstrate in detail how we address the domain heterogeneity problem with the Domain-aware Relative Policy Optimization (DRPO) algorithm and design of appropriate reward functions.

### 3.1  Problem Definition

Each clinical sample is $\mathbf{x}_i = \big(\mathbf{x}_i^{(v)}, \mathbf{x}_i^{(t)}, \mathbf{x}_i^{(s)}, g_i\big)$, where $\mathbf{x}_i^{(v)} \in \mathbb{R}^{P_i \times d_v}$ is a patchified image, $\mathbf{x}_i^{(t)} \in \mathbb{R}^{k_i \times T_i}$ is multichannel time-series data, $\mathbf{x}_i^{(s)}$ is text input, and $g_i \in \{1, \dots, C\}$ denotes the clinical domain (e.g., CT scans, ECG, Chest X-ray). Vision and time-series inputs are optional, which requires the model to handle missing modalities. The learning objectives are to predict: (i) an unsupervisedly learned reasoning trace, (ii) bounding boxes $\mathbf{b}_i = \{b_{i,j}\}_{j=1}^{K_i}$ with $b_{i,j} \in \mathbb{R}^4$ in $(x, y, w, h)$ format highlighting salient image regions, and (iii) a concise diagnosis $\hat{y}_i$.

Table 1: **Comparison of QoQ-Med against other open-source public clinical MLLMs.** BBox: Ability to predict salient bounding boxes; CXR: Chest X-ray; Mammo.: Mammorgraphy; Derm: Dermoscopy; Patho.: Histopathology; US: Ultrasound. BBox: Whether the model is able to produce bounding boxes as output. QoQ-Med is currently the largest medical reasoning MLLM in the field, and the only model trained with DRPO, our RL training algorithm we introduced in Sec. 3.3. * [90, 56] are trained on some ECG images, but none of them are trained on raw ECG time series input.

| Model | Size | Training | BBox | 1D | 2D | | | | | 3D | | |
|---|---|---|---|---|---|---|---|---|---|---|---|---|
| | | | | ECG | CXR | Mammo. | Derm. | Fundus | Patho. | US | MRI | CT |
| **LLaVa-Med** [48] | 7B-13B | SFT | ✓ | ✗ | ✓ | ✗ | ✓ | ✗ | ✓ | ✗ | ✓ | ✓ |
| **Med-Flamingo** [56] | 8.3B | SFT | ✗ | o* | ✓ | ✗ | ✓ | ✗ | ✓ | ✗ | ✓ | ✓ |
| **RadFM** [85] | 14B | SFT | ✓ | ✗ | ✓ | ✓ | ✓ | ✗ | ✗ | ✓ | ✓ | ✓ |
| **BiomedGPT** [90] | 33M-182M | SFT | ✓ | o* | ✓ | ✗ | ✓ | ✓ | ✓ | ✓ | ✓ | ✓ |
| **Med-R1** [46] | 2B | GRPO | ✗ | ✗ | ✓ | ✗ | ✓ | ✓ | ✓ | ✓ | ✓ | ✓ |
| **QoQ-Med (Ours)** | 7B-32B | DRPO | ✓ | ✓ | ✓ | ✓ | ✓ | ✓ | ✓ | ✓ | ✓ | ✓ |

## 3.2 Model

We design the model with an aim that it can take in data across as many domains as possible, so that it can provide comprehensive diagnosis while correlate and co-train across the most diverse range of clinical domains, with inputs ranging from 1D to 3D.

**Model Design.** As shown in Figure 1, we initialize QoQ-Med from a large pretrained vision–language model comprising an image encoder, a linear projection that maps each visual patch embedding into the backbone LLM's token space, and the LLM. To ingest temporal data, we prepend a pretrained time-series encoder, namely ECG-JEPA [43], whose outputs are passed through a newly initialized linear projection of matching dimension. At inference, the projected image patches, time-series tokens, and tokenized text are interleaved in their original temporal order and fed to the LLM. The LLM autoregressively generates a free-text chain of thought, bounding-box tokens that localize the evidence identified in that reasoning, and outputs a short diagnosis. This design supports heterogeneous modality combinations, allowing the model to skip missing channels while preserving positional consistency across the multimodal sequence.

**Training Process.** Training proceeds in two stages. **Stage 1: modality alignment.** Since we initialize the projection layer from scratch, we first train and align the ECG encoder, the projection layer, and the LLM. To encourage high-quality reasoning outputs from the beginning, we use the same DRPO training as in Stage 2. **Stage 2: multimodal fine-tuning with DRPO.** We train on the full multimodal corpus with DRPO, as described in Sec. 3.3, which balances training across different samples in various domains and difficulty. In this stage, we aim to simultaneously improve the diagnostic accuracy and reasoning quality, with rewards described in Sec. 3.4.

**Training Data.** We train the unified vision and time-series model across 33 datasets using the CLIMB dataset [22]. The dataset contains 2.61 million samples across 1D (ECG), 2D (Chest X-ray, Mammography, Dermoscopy, histopathology, Fundus), and 3D (Ultrasound, MRI, CT Scan) data. The exact composition of the data and the training hyperparameters are included in App. C and D.

**Comparison with current public clinical MLLMs.** Table 1 demonstrates that our model is currently the largest open clinical MLLM in the field. It is also the only model that can both take in time series data and output its thinking process, along with the bounding box annotation highlighting the salient region made during the thinking process.

## 3.3 Domain-aware Relative Policy Optimization (DRPO)

Group Relative Policy Optimization (GRPO) is a reinforcement learning method that gained prominence following the success of DeepSeek-R1. Unlike Proximal Policy Optimization (PPO), which relies on a separate value network to estimate advantages, GRPO directly computes the advantage $\hat{A}_{(q,i,t)}$ (Eq. 1) for each response within a group of rollouts $G_{(q,t)}$ at a given training iteration. A rollout refers to a single sampled trajectory or response generated by the policy in reaction to a prompt. The advantage quantifies how much better a particular rollout is compared to others generated for the same prompt, enabling the policy to prioritize relatively high-quality responses without requiring an explicit estimate of expected return.

Each group of rollouts $G_{(q,t)}$ consists of multiple responses sampled for the same prompt $q$. Let $r_{(q,i,t)}$ denote the scalar reward assigned to the $i$-th response $o_{(q,i,t)}$ at time step $t$, where

each response is a sequence of tokens $o_{(q,i,t)} := o_{(q,i,t):1}, o_{(q,i,t):2}, \ldots, o_{(q,i,t):n_{o_{(q,i,t)}}}$, and $n_{o_{(q,i,t)}}$ denotes the length of the token sequence. The set of rewards for the group is defined as $R_{G_{(q,t)}} = \{r_{(q,1,t)}, r_{(q,2,t)}, \ldots, r_{(q,|G_q|,t)}\}$, where $|G_q|$ is the number of responses in the group. GRPO normalizes these rewards to have zero mean and unit variance, producing the normalized advantage:

$$\hat{A}^{\text{GRPO}}_{(q,i,t)} = \frac{r_{(q,i,t)} - \hat{\mu}_{G_{(q,t)}}}{\hat{\sigma}_{G_{(q,t)}} + \varepsilon}, \tag{1}$$

where $\hat{\mu}_{G_{(q,t)}}$ and $\hat{\sigma}_{G_{(q,t)}}$ denote the empirical mean and standard deviation of the group rewards, respectively, and $\varepsilon$ is a small constant added for numerical stability. These advantage estimates are incorporated into the GRPO clipped surrogate objective, which also includes a per-token KL divergence penalty:

$$\tilde{A}_{(q,i,t):k}(\theta) = \min\left(\varphi_{(q,i,t):k}(\theta) \cdot \hat{A}^{\text{GRPO}}_{(q,i,t)}, \ \text{clip}\left(\varphi_{(q,i,t):k}(\theta), \ 1-\varepsilon, \ 1+\varepsilon\right) \cdot \hat{A}^{\text{GRPO}}_{(q,i,t)}\right),$$

$$\varphi_{(q,i,t):k}(\theta) = \frac{\pi_\theta(o_{(q,i,t):k} \mid q, o_{(q,i,t):<k})}{\pi_{\theta_{\text{old}}}(o_{(q,i,t):k} \mid q, o_{(q,i,t):<k})},$$

$$J_{\text{GRPO}}(\theta) = \mathbb{E}_{q \sim \mathcal{D}, \{o_{(q,i,t)}\} \sim \pi_{\theta_{\text{old}}}} \left[ \frac{1}{|G_{(q,t)}|} \sum_{i=1}^{|G_{(q,t)}|} \frac{1}{n_{o_{(q,i,t)}}} \sum_{k=1}^{n_{o_{(q,i,t)}}} \tilde{A}_{(q,i,t):k}(\theta) - \beta \, D_{\text{KL}}\left(\pi_\theta \,\|\, \pi_{\text{ref}}\right) \right].$$

Here, $o_{(q,i,t):<k}$ refers to the token subsequence from position 1 to $k-1$, and $\mathcal{D}$ denotes the dataset distribution. The term $\varphi_{(q,i,t):k}(\theta)$ represents the importance sampling ratio between the current policy $\pi_\theta$ and the old policy $\pi_{\theta_{\text{old}}}$ at token position $k$; $\hat{A}^{\text{GRPO}}_{(q,i,t)}$ is the normalized advantage estimate for the $i$-th response in group $G_{(q,t)}$; $\varepsilon$ is a small constant used for numerical stability and clipping; $\beta$ is a scalar hyperparameter that controls the strength of the KL divergence regularization; and $D_{\text{KL}}(\pi_\theta \| \pi_{\text{ref}})$ denotes the Kullback–Leibler divergence between the learned policy and a reference policy. GRPO demonstrates strong empirical performance when the input data is relatively homogeneous. However, in settings with high data heterogeneity, domains with abundant samples tend to dominate the optimization process, while under-represented domains contribute minimally. This imbalance can bias the model and degrade performance on rare but clinically important modalities, while spending too much compute on easy problems on abundant domains.

**Domain-aware Relative Policy Optimization (DRPO).** While GRPO normalizes reward signals across rollouts that respond to the *same* prompt—thereby reducing variance within a group and ensuring fairer comparison among responses—it does not address imbalance *across* domains. As a result, domains that appear more frequently in the training data continue to have a disproportionate impact on the learning process. DRPO builds on GRPO by introducing a hierarchical scaling mechanism that explicitly balances contributions from different domains. This correction for inter-domain imbalance preserves GRPO's simplicity and value-free formulation while promoting more equitable learning across heterogeneous data distributions.

**Hierarchical Cluster-Based Scaling.** The core innovation of DRPO lies in a hierarchical scaling strategy that adaptively balances learning signals based on both domain frequency and task difficulty. This mechanism operates at two levels: across domains, to mitigate the dominance of overrepresented domains, and within domains, to adjust for variations in response quality or reward magnitude. Concretely, we first cluster question-level reward sets within each domain, treating each set of individual rewards as a feature vector. We then apply a two-stage reward scaling procedure—first at the cluster level, then at the individual reward level—thereby emphasizing learning from rare and challenging questions.

**Stage-1: Intra-Domain Clustering.** At each iteration step $t$, we begin by sampling an independent batch of questions. These questions are then clustered into different domains. Let $g$ denote a domain, and let $N_{(g,t)}$ represent the number of questions in domain $g$ at iteration $t$. Within each domain at iteration $t$, we first compute the set of rewards for each question. These rewards, collected across multiple rollouts, are concatenated into a feature vector per question. Specifically, for each domain $g$, we construct a set of reward vectors $\mathcal{H}_g = \{\mathbf{v}_q^g\}_{q=1}^{N_g}, \quad \mathbf{v}_q^g \in \mathbb{R}^{|G_{(q,t)}|}$, where $\mathbf{v}_q^g$ contains the $R_{G_{(q,t)}}$ rollout rewards for question $q$, and $N_{(g,t)}$ is the number of questions in domain $g$, at iteration step t.

To uncover patterns in question difficulty, we apply K-means clustering to these reward vectors at each time step $t$, separately within each domain:

$$\{\mathbf{C}_{(1,g,t)}, \mathbf{C}_{(2,g,t)}, \ldots, \mathbf{C}_{(k_{(g,t)},g,t)}\} = \text{KMeans}(\mathcal{H}_g, k_{(g,t)}),$$

where $\mathbf{C}_{(c,g,t)}$ denotes the centroid of cluster $c$ in domain $g$, and $k_{(g,t)}$ is the number of clusters, which is determined automatically using the elbow method (see Appendix B.1).

**Stage-2: Hierarchical Scaling.** For each domain and each cluster within that domain, we compute **inter-domain** temperature factors $T_{(g,t)}$ and **intra-domain** temperature factors $T_{(c,g,t)}$. These factors capture both the relative size and average difficulty of each domain and cluster. Difficulty is estimated using the mean reward, either per domain or per cluster within the domain, which serves as a proxy for how easy or challenging the questions are within each specific domain and cluster. These temperature factors are then *inversely multiplied* with the corresponding advantage functions—at both the domain and cluster levels—so that domains and clusters that are smaller or harder receive proportionally greater weight during training. Concretely:

$$T_{(g,t)} = \max\left(\sqrt{N_{(g,t)}} \cdot \mu_{(g,t)}, \varepsilon\right), \quad T_{(c,g,t)} = \max\left(\sqrt{N_{(c,g,t)}} \cdot \mu_{(c,g,t)}, \varepsilon\right), \tag{2}$$

where $N_{(c,g,t)}$ is the size of cluster $c$, and $\mu_{(g,t)}$ and $\mu_{(c,g,t)}$ denote the mean reward for group $g$ and cluster $c$ in group $g$, at iteration $t$.

To scale reward advantage with the appropriate temperature factors, we first normalize rewards at the question level as in GRPO, then scale by the domain and cluster temperatures, before multiplying by a KL regularization factor $m_{(i,t)}$. Concretely,

$$s_{(q,i,t)}^{scaled} = \frac{m_{(i,t)} \cdot s_{(q,i,t)}}{T_{(g,t)} \cdot T_{(c,g,t)}}, \tag{3}$$

where $s_i = \frac{r_{i,t} - \mu_{q,t}}{\sigma_q + \varepsilon}$ is the question level-normalized reward from GRPO. The KL regularization is applied to prevent outliers from dominating the update, as detailed in Appendix B.2. Finally, we scale the standard deviation back to 1 by dividing each reward by the standard deviation of the reward in the batch $\hat{A}_{(q,i,t)}^{\text{DRPO}} = \frac{s_{(q,i,t)}^{scaled}}{\sigma_{s_t^{scaled}}}$.

**DRPO Objective.** DRPO maintains the same objective structure as GRPO, maximizing:

$$\tilde{A}_{(q,i,t):k}(\theta) = \min\left(\varphi_{(q,i,t):k}(\theta) \cdot \hat{A}_{(q,i,t)}^{\text{DRPO}}, \text{ clip}\left(\varphi_{(q,i,t):k}(\theta), 1-\varepsilon, 1+\varepsilon\right) \cdot \hat{A}_{(q,i,t)}^{\text{DRPO}}\right),$$

$$J_{\text{DRPO}}(\theta) = \mathbb{E}_{q \sim \mathcal{D}, \{o_{(q,i,t)}\} \sim \pi_{\theta_{\text{old}}}}\left[\frac{1}{|G_{(q,t)}|} \sum_{i=1}^{|G_{(q,t)}|} \frac{1}{n_{o_{(q,i,t)}}} \sum_{k=1}^{n_{o_{(q,i,t)}}} \tilde{A}_{(q,i,t):k}(\theta) - \beta\, D_{\text{KL}}\left(\pi_\theta \,\|\, \pi_{\text{ref}}\right)\right],$$

where $\varphi_{(q,i,t):k}(\theta) = \frac{\pi_\theta(o_{(q,i,t):k}|q, o_{(q,i,t):<k})}{\pi_{\theta_{\text{old}}}(o_{(q,i,t):k}|q, o_{(q,i,t):<k})}$.

**Benefits of DRPO.** The cluster-based DRPO approach offers several key benefits:

1. **Hierarchical Scaling:** DRPO implements two-layer scaling: first at the domain level and then at the cluster level within each domain. This directs optimization toward both underrepresented domains and challenging question subsets, ensuring the model learns effectively across all data types. This approach prevents the model from focusing only on easy or common problems while neglecting rare but important clinical scenarios.
2. **Preservation of Zero Mean and Unit Variance:** DRPO scales rewards after GRPO normalization, maintaining the property that the mean reward within each set of rollouts remains 0 and the standard deviation is 1. This property is crucial for stable optimization in reinforcement learning, as established in previous works [14, 94, 57].
3. **Computational Efficiency:** DRPO operates with minimal additional complexity of order $O(n)$, primarily from the K-means algorithm operating on low-dimensional vectors (typically 5-10 elements). This enables efficient training without the overhead of critic networks, making it particularly suitable for large-scale LLM fine-tuning.

## 3.4 Reward Design

During the training of QoQ-Med, we employ a combination of two main rewards and two auxiliary rewards that balance diagnostic accuracy with interpretability, a critical requirement for clinical applications where understanding model reasoning.

**Accuracy reward.** The primary goal of our model is diagnostic accuracy, for which we compute a standard *accuracy reward* $r_i^{\mathrm{acc}}$. We treat prediction $\hat{y}_i$ and ground truth $y_i$ as unordered sets of labels and assign $r_i^{\mathrm{acc}} = \mathrm{F1}(\hat{y}_i, y_i)$, which directly optimizes the model's ability to identify correct diagnoses across diverse clinical scenarios.

**Semantic alignment reward.** For clinical applications, the ability to identify and highlight relevant regions in medical imagery is crucial for building clinician trust. The *semantic alignment reward* encourages the model to correctly identify salient regions that support its diagnostic decisions. Let $\mathbf{b}_i = \{b_{i,j}\}_{j=1}^{K_i}$ be the set of axis-aligned bounding boxes output by the model and $S_i \subseteq [0,1]^{H \times W}$ the pixel-level segmentation mask associated with the ground-truth diagnosis. We define this reward as the best intersection-over-union score: $r_i^{\mathrm{IoU}} = \max_{j=1,\ldots,K_i} \frac{\mathrm{area}\left(b_{i,j} \cap S_i\right)}{\mathrm{area}\left(b_{i,j} \cup S_i\right)}$. By optimizing this reward, the model learns to visually highlight the specific anatomical regions relevant to its diagnosis, providing critical interpretability for clinical decision support.

**Auxiliary rewards.** We also employ auxiliary rewards that encourage proper formatting and comprehensive reasoning, detailed in Appendix B.3. These rewards help ensure that the model's outputs are well-structured and sufficiently detailed for clinical use.

**Combined reward.** The final scalar reward supplied to DRPO is a weighted combination: $r_i = \lambda_{\mathrm{acc}} r_i^{\mathrm{acc}} + \lambda_{\mathrm{IoU}} r_i^{\mathrm{IoU}} + \lambda_{\mathrm{aux}} r_i^{\mathrm{aux}}$. In our experiments, we set $(\lambda_{\mathrm{acc}}, \lambda_{\mathrm{IoU}}, \lambda_{\mathrm{aux}}) = (0.6, 0.2, 0.2)$.

# 4 Experiments

We design experiments to answer the following research questions. Details are included in App. D.

**RQ1: How does DRPO compare with other critic-free RL methods and models?** As detailed in Sec. 3.2, we train and evaluate QoQ-Med on a combination of 30 clinical diagnosis datasets across 9 clinical domains. A description of each dataset is included in App. C. The models are evaluated with balanced accuracy and macro-F1. We compare our training method DRPO against supervised fine-tuning (SFT), PPO [74] and four popular critic-free RL training methods: GRPO [75], RLOO [2], Reinforce++ [33], and ReMax [50]. We further compare our trained model QoQ-Med against medical VLMs (Llava-Med [48], Med-R1 [46]) and closed source VLMs (GPT-4o [37], o4-mini [61]).

**RQ2: How well does DRPO handle mixed multimodal inputs?** We repeat the comparison on MIMIC-IV, where samples contain a chest X-ray, a 12-lead ECG trace, and an accompanying clinical record. We train and evaluate the models on two tasks: length of stay (LOS) prediction, binned into a 4-day interval, and 48-hour in-hospital mortality (48-IHM). We evaluate the model with accuracy and F1 score in the same way as RQ1.

**RQ3: How is the quality of the reasoning traces and bounding boxes learned by DRPO?** We did both a qualitative and a quantitative analysis on QoQ-Med's reasoning and bounding box outputs. We evaluate the bounding box quality via the intersection over union (IoU) against the ground truth segmentation available in the dataset. We further collaborated with clinicians to annotate the reasoning traces on the validation dataset, grading the traces by their relevance to the final diagnosis.

## 4.1 RQ1: Comparison with other RL Training Methods and Models

**Comparison with other RL methods.** Table 2 shows a comparison between DRPO and several critic-free RL training methods across eight medical imaging modalities. The results demonstrate that DRPO consistently outperforms all competing methods in 6 out of 8 vision modalities in terms of F1 score. Overall, DRPO achieves a mean accuracy that is 5.9% higher in percentage points and an F1 score that is 46% higher compared to the best critic-free baseline method. As compared to GRPO in Fig. 2(a), the most substantial increase is observed in datasets from understudied modalities, like ultrasound and mammography, as defined in App. C.2. As shown in Fig. 2(b), QoQ-Med achieves the best performance across all clinical domains as compared to current open-source MLLMs. Compared

Table 2: **Performance comparison of medical vision-language models across various medical imaging modalities.** Acc: Accuracy, F1: F1 Score, CXR: Chest X-ray. DRPO training outperforms various other RL training methods and SFT across 7 out of 8 medical imaging domains in F1 score. The metrics are averaged across four separate training runs. Metrics with standard deviation is included in App. Tab. 7.

| Model | CXR | | Mammo. | | Dermoscopy | | CT Scan | | Fundus | | Ultrasound | | MRI | | Pathology | | Overall | |
|---|---|---|---|---|---|---|---|---|---|---|---|---|---|---|---|---|---|---|
| | Acc | F1 | Acc | F1 | Acc | F1 | Acc | F1 | Acc | F1 | Acc | F1 | Acc | F1 | Acc | F1 | Acc | F1 |
| **SFT** | .688 | .078 | .481 | .056 | .640 | .158 | .525 | .236 | **.715** | .066 | .548 | **.235** | .567 | .197 | .652 | .083 | .602 | .139 |
| **PPO** [74] | .670 | .064 | .738 | .205 | .668 | .278 | .571 | .257 | .669 | .083 | .490 | .080 | .767 | .540 | .745 | .364 | .665 | .234 |
| **ReMax** [50] | .636 | .120 | .577 | .033 | .644 | .257 | .567 | .228 | .678 | .089 | .547 | .147 | .547 | .264 | .706 | .270 | .596 | .176 |
| **RE++** [33] | .730 | .082 | .660 | .076 | .635 | .237 | .529 | .247 | .672 | .098 | .519 | .136 | .651 | .420 | .668 | .254 | .621 | .202 |
| **RLOO** [2] | **.752** | .086 | .471 | .068 | .636 | .216 | .534 | .224 | .670 | **.099** | .519 | .144 | .658 | .432 | .699 | .216 | .611 | .189 |
| **GRPO** [75] | .703 | .095 | .466 | .059 | .646 | .244 | .524 | .236 | .670 | .086 | .520 | .146 | .631 | .395 | .715 | .286 | .609 | .193 |
| **DRPO**$_{\text{DomainOnly}}$ | .693 | .086 | .751 | .213 | .679 | .251 | .571 | .257 | .669 | .083 | .480 | .098 | .733 | .475 | **.762** | **.388** | **.668** | .237 |
| **DRPO**$_{\text{NoKL}}$ | .685 | .103 | .711 | **.264** | .691 | .382 | **.597** | **.365** | .676 | .085 | .554 | .228 | .722 | .535 | .710 | .300 | **.668** | .283 |
| **DRPO** | .687 | **.115** | **.756** | .253 | **.715** | **.407** | .570 | .309 | .672 | .093 | **.555** | .223 | **.789** | **.625** | .708 | .265 | .666 | **.295** |

Table 3: **Ablation studies on cluster size and reward composition.** Acc: Accuracy, F1: F1 Score. Bold values indicate best performance within each ablation group.

| Config | CXR | | Mammo. | | Dermoscopy | | CT Scan | | Fundus | | Ultrasound | | MRI | | Pathology | | Overall | |
|---|---|---|---|---|---|---|---|---|---|---|---|---|---|---|---|---|---|---|
| | Acc | F1 | Acc | F1 | Acc | F1 | Acc | F1 | Acc | F1 | Acc | F1 | Acc | F1 | Acc | F1 | Acc | F1 |
| *Cluster Size* | | | | | | | | | | | | | | | | | | |
| **1** | **.694** | .085 | .746 | .211 | .678 | .286 | .571 | .257 | .669 | .083 | .544 | .200 | .757 | .505 | **.773** | **.449** | .679 | .259 |
| **3** | **.694** | .125 | .568 | .048 | .680 | .356 | .562 | .284 | **.672** | .147 | .520 | .152 | .717 | .546 | .723 | .289 | .642 | .244 |
| **10** | .691 | .125 | **.759** | .253 | **.707** | **.400** | **.580** | **.321** | .670 | .088 | **.568** | **.240** | **.806** | **.652** | .707 | .303 | **.686** | .286 |
| **20** | .668 | **.167** | .751 | **.268** | .675 | .300 | .548 | .262 | .635 | **.166** | .547 | .214 | .804 | .649 | .731 | .329 | .670 | **.294** |
| *Reward Composition (Acc:IoU)* | | | | | | | | | | | | | | | | | | |
| **0.6:0.2** | .691 | .125 | **.759** | **.253** | **.707** | **.400** | **.580** | **.321** | .670 | .088 | .568 | **.240** | **.806** | **.652** | **.707** | **.303** | **.686** | **.286** |
| **0.2:0.6** | .690 | **.147** | .563 | .185 | .668 | .290 | .576 | .308 | **.681** | **.136** | **.573** | .218 | .768 | .561 | .698 | .233 | .652 | .260 |

to the closed-source commercial models, it achieves the best performance against GPT-4o [37], while surpassing the reasoning model GPT-o4-mini [60] in all domains except MRI.

**Ablations.** The substantial improvement in F1 score can be attributed to two key components of DRPO. First, the introduction of domain-wise scaling contributes to a significant 22.8% improvement in F1 score, as evidenced by the performance difference between DRPO$_{\text{DomainOnly}}$ and vanilla GRPO. Subsequently, after incorporating clustering within each domain and specifically encouraging the model to focus on small, challenging clusters within each domain, the performance is further enhanced by an additional 19.4% in terms of F1 score.

Tab. 3 shows further ablations on the number of clusters and reward compositions. In general, we found that the weight of each reward does not have a significant impact on the final performance. In particular, the auxiliary rewards on formatting saturate shortly in the early stages of training. They have effectively no impact on the later stages due to normalization. We tested different combinations of accuracy rewards: semantic alignment rewards. As demonstrated in the table, decreasing the weight of the accuracy reward gives a drop in overall performance and performance in most domains, but results are still significantly better than all baselines, which demonstrates the robustness of DRPO.

The number of clusters in the model is determined automatically via the elbow method, with the possibility to set an upper limit on the number of clusters. As a part of the ablation, we tested the model with 1 (no clustering), 3, 10 and 20 clusters, and included the results in Tab. 3. In general, we observe that having no cluster or a very low cluster limit will cause a decrease in performance. A higher cluster limit, however, does not seem to hurt the performance, as the elbow method automatically chooses a lower cluster count than the limit. This allows the algorithm to remain efficient under arbitrary cluster limits.

**Runtime Efficiency.** As shown in Fig. 3(c), while DRPO requires clusters to be calculated on each step, it has a negligible impact on the overall runtime. Across all critic-free RL methods, reward calculation accounts for less than 2% of the total runtime of a step.

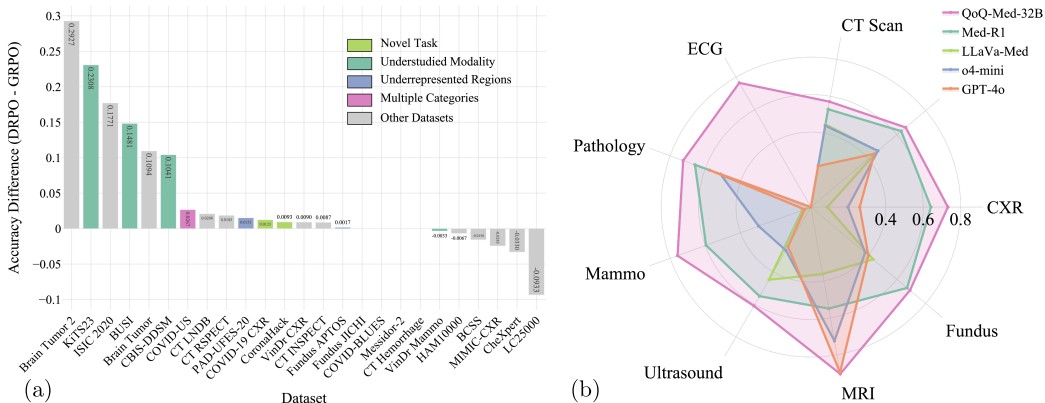

Figure 2: **(a) Difference in accuracy (DRPO - GRPO).** DRPO brings the most performance gain in understudied modalities as defined in App. C.2. **(b) Accuracy comparison of QoQ-Med against SoTA open source and closed source LLMs.** QoQ-Med outperforms all open and closed MLLMs across 8 domains. The full results are included in App. Table 5.

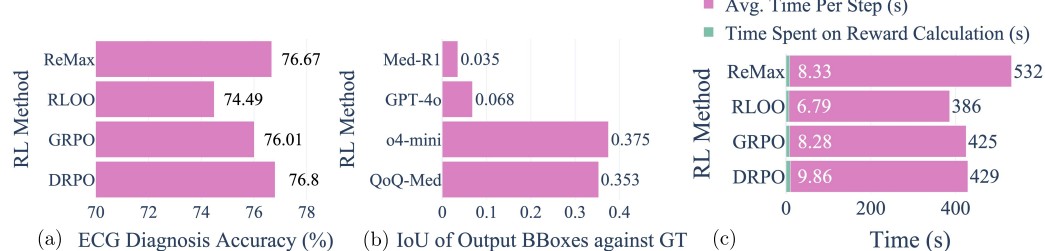

Figure 3: **(a) Accuracy of ECG Diagnosis.** DRPO models reach the best performance among all critic-free RL methods. **(b) Intersection over Union (IoU)** of model-generated bounding boxes against truth labels. QoQ-Med (Ours) surpasses open source models and has a performance on par with o4-mini. **(c) Per Step Runtime** of reward calculation of RL methods on 8xA100 GPUs. While DRPO adds hierarchical clustering, the runtime of the reward calculation still accounts for less than 2% of the total runtime per step and has minimal impact on training.

## 4.2 RQ2: Multimodal Fusion Performance

We tested how the model integrates multiple modalities and how much each modality contributes to the final diagnostic accuracy via MIMIC-IV [41] dataset. On the MIMIC-IV dataset, the model has to reason across ECGs, chest X-rays, and health records. As shown in Tab. 4, we found DRPO allows the model to reach a better performance in both tasks as compared to GRPO. In addition, taking full inputs across ECG, Chest X-ray images, and electronic health records (EHR) gives better performance than any ablation of these modalities, signaling that QoQ-Med is able to effectively aggregate information across all modalities. Specifically, we found vision and texts contirbute more to the

Table 4: **Models' Perf. on MIMIC-IV.** DRPO-Full with inputs from 3 modalities has the best performance for both tasks, time-series only (DRPO-TS+T) and vision only (DRPO-Vision+T) ablations having worse performance, and the text only ablation having the worst performance.

| Algo/Inputs | LOS | | 48-IHM | |
|---|---|---|---|---|
| | Acc | F1 | Acc | F1 |
| **GRPO-Full** | 0.626 | 0.105 | 0.551 | 0.354 |
| **DRPO-TextOnly** | 0.645 | 0.195 | 0.563 | 0.583 |
| **DRPO-ECG+T** | 0.639 | 0.204 | 0.602 | 0.528 |
| **DRPO-Vision+T** | **0.669** | 0.223 | 0.596 | 0.586 |
| **DRPO-Full** | 0.663 | **0.283** | **0.642** | **0.597** |

final accuracy and F1 scores than ECG. While QoQ-Med represents a first step towards multimodal reasoning models across vision and time series, future works could explore better architecture, data, or training methods that better balances the power of each modalities.

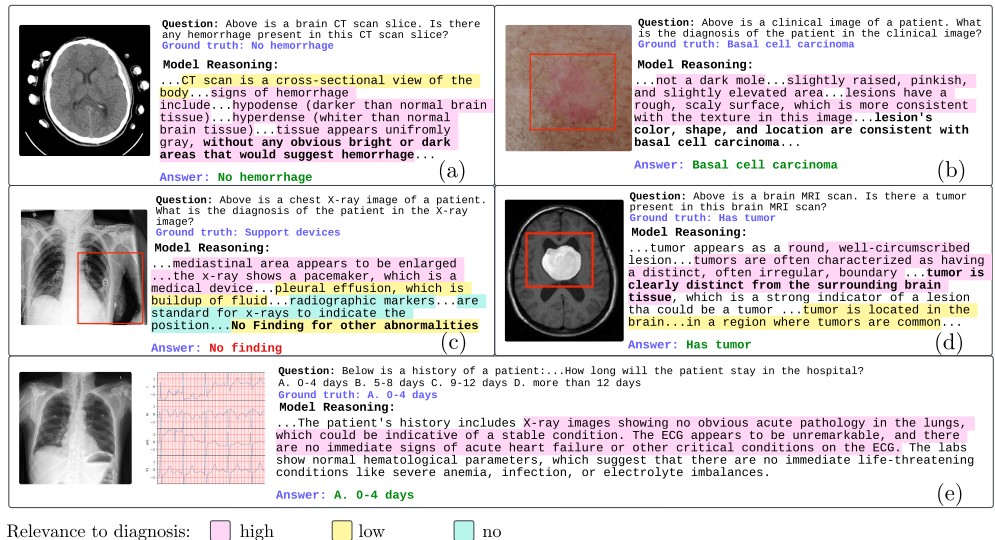

Relevance to diagnosis: ☐ high ☐ low ☐ no

Figure 4: **Model outputs annotated by clinical experts.** QoQ-Med correctly reasons from modality-specific clinical knowledge, generates bounding boxes, and outputs the correct predictions in most instances except (c). (e) demonstrates the model's ability to synthesize multimodal inputs with reasoning. The bounding boxes correctly highlight the salient regions related to the reasoning steps when one is present.

## 4.3 RQ3: Quality of Reasoning Traces

**Clinician relevance annotations.** App. D.4.2 provides a breakdown of clinician-annotated reasoning traces, revealing that the model mostly generates contents highly relevant to the diagnosis, with minimal output judged as irrelevant. We observe that the model often correctly recalls relevant clinical knowledge, which help guide the model by providing associative context. For example, in Fig 4(a), the model correctly recalls different signs of hemorrhage on CT, such as darker or whiter tissues, and relates this context to specific parts of the image to make a correct prediction. In Fig. 4(c), the model correctly identifies the presence of a pacemaker, indicating a support device, but subsequently concludes that there are no additional abnormalities, ultimately leading it to predict "No finding". This suggests that while the model's final predictions may be incorrect, its intermediate reasoning often reflects clinically relevant patterns.

**Bounding box quality.** Fig. 3(b) demonstrates that the model identifies bounding boxes correlated with the ground truth annotations, with the IoU exceeding the best open source models while reaching a similar performance as the closed-source reasoning model o4-mini. From Fig. 4, we also see that the outputs by the model are sufficiently aligned with the reasoning process, allowing the clinicians to confirm the model's predictions while cross-referencing the source image.

## 5 Conclusion

We introduced QoQ-Med, a clinical MLLM with reasoning across 9 clinical domains. Our Domain-aware Group Relative Policy Optimization (DRPO) demonstrates superior performance over existing approaches, with up to 43% improvement in average F1 score across clinical modalities and substantial gains in multimodal fusion tasks. The ability of QoQ-Med to process 1D time series data alongside traditional 2D/3D clinical images addresses a significant gap in existing medical multimodal systems, while its transparent reasoning process enhances interpretability and clinical trust. By publicly releasing QoQ-Med-7B/32B and our comprehensive reasoning dataset containing 2.61 million question-answer pairs, we hope to contribute valuable resources to advance clinical reasoning AI. A potential limitation is the limited sample efficiency as the reasoning process is not supervised. Moving forward, we hope the community can explore ways to elicit high-quality reasoning with better data efficiency, with a special focus on understudied modalities like ECG and ultrasound.

# 6 Acknowledgement

This material is based upon work supported by the National Science Foundation Graduate Research Fellowship under Grant No. 2141064. Any opinion, findings, and conclusions or recommendations expressed in this material are those of the authors and do not necessarily reflect the views of the National Science Foundation.

We thank the MIT Office of Research Computing and Data (ORCD) for support through ORCD Seed Fund Grants, which provided access to 8xH200 GPUs and additional funding support. We also thank the NVIDIA Academic Grant Program for GPU support.

We also extend our sincere thanks to Haowen Wei (Research Associate, MIT Institute for Medical Engineering & Science, MIT.nano) and Dr. Farzan Vahedifard (Neurologist, Athinoula A. Martinos Center for Biomedical Imaging, Harvard Medical School) for their careful annotation of the model's reasoning outputs and valuable insights that significantly improved this work.

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

# A  Impact Statement

This paper presents the development, training, and evaluation of QoQ-Med, a multimodal language model for clinical reasoning. QoQ-Med is the first foundational model to jointly process and reason over medical images, ECG signals, and clinical text. By incorporating underused physiological signals such as ECG, our model moves beyond vision-centric approaches and supports broader applications in cardiology. These robust models can then assist clinicians in interpreting diverse forms of patient data, identifying patterns, and generating more complete diagnostic assessments.

As multimodal health care models become increasingly used, transferability and accountability become the focus. It is no longer sufficient for models to produce accurate predictions, but also to justify how those predictions were made. By incorporating a reasoning component, we aim to build trust and facilitate broader deployment of healthcare AIs in real-world settings.

To address data and privacy concerns, we have taken appropriate steps to access consented data. We have anonymized sensitive personal information such as gender and race from the datasets we use.

The model is mainly oriented for clinical use. However, before extensive real world testings (like human trials), the model is not suitable for real clinical deployment. We address this during the download of model weights, emphasizing this is a research review, not a product approved by federal agencies.

# B  DRPO Implementation Details

## B.1  Elbow Method for K-means Cluster Selection

To automatically determine the optimal number of clusters $k_{\text{opt}}$ for each domain, we implement the elbow method that balances clustering granularity against overfitting. For each domain's question vectors, we:

1. Compute the inertia (sum of squared distances to assigned centroids) for a range of cluster counts $k \in \{1, 2, \ldots, k_{\text{max}}\}$, where $k_{\text{max}} = \min(10, N_{\text{unique}})$ and $N_{\text{unique}}$ is the number of unique question vectors.
2. Calculate the decrease in inertia $\Delta I_k = I_{k-1} - I_k$ for each additional cluster.
3. Identify the "elbow point" where adding more clusters yields diminishing returns, specifically where $\Delta I_k < \tau \cdot \Delta I_{k-1}$ for a tolerance parameter $\tau = 0.10$.

This approach prevents excessive fragmentation of the question space while still capturing meaningful patterns in question difficulty. In practice, this typically results in a small number of clusters (2-5) per domain, which is computationally efficient while capturing the primary patterns in question difficulty.

## B.2  KL-aware Regularization

The hierarchical scaling in DRPO can increase the variance of advantages, potentially leading to optimization instability where outliers dominate updates. To address this, we implement KL-aware regularization that dampens a small percentage of advantages proportionally to their KL divergence from the reference model.

For each response $o_{q,i,t}$, we compute a question-level KL divergence:

$$k_{(q,t)} = \sum_i \left[ \pi_\theta \log \frac{\pi_\theta}{\pi_{\text{ref}}} \right]_{q,t} \tag{4}$$

We then apply an inverse-linear regularizer to the scaled advantages:

$$m_{(q,i,t)} = \frac{t_p}{t_p + \max(s_{(q,i,t)}^{scaled} \cdot k_{(q,t)}, 0)} \tag{5}$$

where $t_p$ is the $p$-th percentile of the values $\{s_i^{scaled} \cdot k_i\}$ within the mini-batch, and $p = 0.9$ in our experiments.

Table 5: **Comparison with other general domain and medical MLLMs.** Acc: Accuracy, F1: F1 Score. QoQ-Med-7B is the best model among open-source clinical MLLMs across all categories. It rivals closed source models in terms of accuracy but has a lower F1 score.

| Model | Chest X-Ray | | Mammo. | | Dermoscopy | | CT Scan | | Fundus | | Ultrasound | | MRI | | Pathology | | Overall | |
|---|---|---|---|---|---|---|---|---|---|---|---|---|---|---|---|---|---|---|
| | Acc | F1 | Acc | F1 | Acc | F1 | Acc | F1 | Acc | F1 | Acc | F1 | Acc | F1 | Acc | F1 | Acc | F1 |
| **Open Source Clinical VLMs** | | | | | | | | | | | | | | | | | | |
| **LLaVa-Med** | .088 | - | .049 | - | .466 | - | .448 | - | .434 | - | .448 | - | .363 | - | .000 | - | .287 | - |
| **Med-R1** | .641 | .002 | .596 | .029 | .630 | .000 | .530 | .007 | **.671** | .062 | .549 | .008 | .550 | .000 | .659 | .000 | .603 | .013 |
| **Closed Source Commercial VLMs** | | | | | | | | | | | | | | | | | | |
| **o4-mini** | .198 | .189 | .297 | .271 | .467 | .445 | .441 | .389 | .378 | .354 | .267 | .205 | .725 | .775 | .514 | .522 | .411 | .394 |
| **GPT-4o** | .261 | .244 | .036 | .056 | .442 | .383 | .222 | .228 | .401 | .299 | .244 | .231 | .896 | .908 | .575 | .565 | .385 | .364 |
| **QoQ-Med-7B** | **.691** | **.125** | **.759** | **.253** | **.707** | **.400** | **.580** | **.321** | .670 | **.088** | **.568** | **.240** | **.806** | **.652** | .707 | **.303** | **.686** | **.286** |
| **QoQ-Med-32B** | **.732** | **.203** | **.758** | **.224** | **.661** | **.290** | **.571** | **.257** | **.691** | **.148** | **.609** | **.260** | **.904** | **.810** | **.726** | **.336** | **.707** | **.316** |

This regularization has several advantages:

1. It selectively dampens responses with both high advantage and high KL divergence, which are likely to represent overconfident "shortcuts" rather than genuine improvements.
2. It requires no additional computation since the KL divergence $k_i$ is already calculated for the main DRPO objective.
3. It preserves most advantages while providing robustness against spurious outliers.

### B.3  Auxiliary Rewards

In addition to the primary accuracy and semantic alignment rewards described in Section 3.4, we implement auxiliary rewards to enhance the overall quality and completeness of the model's outputs.

**Format reward.**  Clinical diagnoses require structured and complete explanations. To ensure the model provides comprehensive evidence for each diagnosis, we implement a *format reward*:

$$r_i^{\text{fmt}} = \begin{cases} 1 & \text{if a bounding box is present for } every \text{ predicted label,} \\ 0 & \text{otherwise.} \end{cases} \tag{6}$$

This binary reward penalizes responses that make claims without providing visual evidence, encouraging the model to explicitly support each diagnostic conclusion with localized visual evidence. This is particularly important in clinical settings where unsubstantiated diagnoses can lead to improper treatment decisions.

**Visual coherence.**  When the dataset includes multiple images or views, we may optionally include a visual coherence reward that encourages consistency in identified regions across different views of the same structure. This is defined as:

$$r_i^{\text{coh}} = \frac{1}{|V_i|} \sum_{v \in V_i} \text{IoU}(b_{i,v}, \mathcal{T}(b_{i,v'})) \tag{7}$$

where $V_i$ is the set of views, $b_{i,v}$ is the bounding box in view $v$, and $\mathcal{T}$ is a transformation function that maps between coordinate systems of different views. This reward is particularly valuable for modalities like echocardiography where multiple views of the same structure are available.

These auxiliary rewards comprise a small portion of the total reward signal but play an important role in shaping the model's outputs toward clinically useful formats. In practice, we found that even with small weights, these auxiliary rewards significantly improve the practical usability of model outputs as judged by clinical evaluators.

## C  Details of the Datasets used in Training and Validation

We use the CLIMB dataset to train our QoQ-Med model. CLIMB is a multimodal clinical diagnosis dataset introduced in Dai et al. [22]. It contains a mixture of 44 publicly available datasets across 13

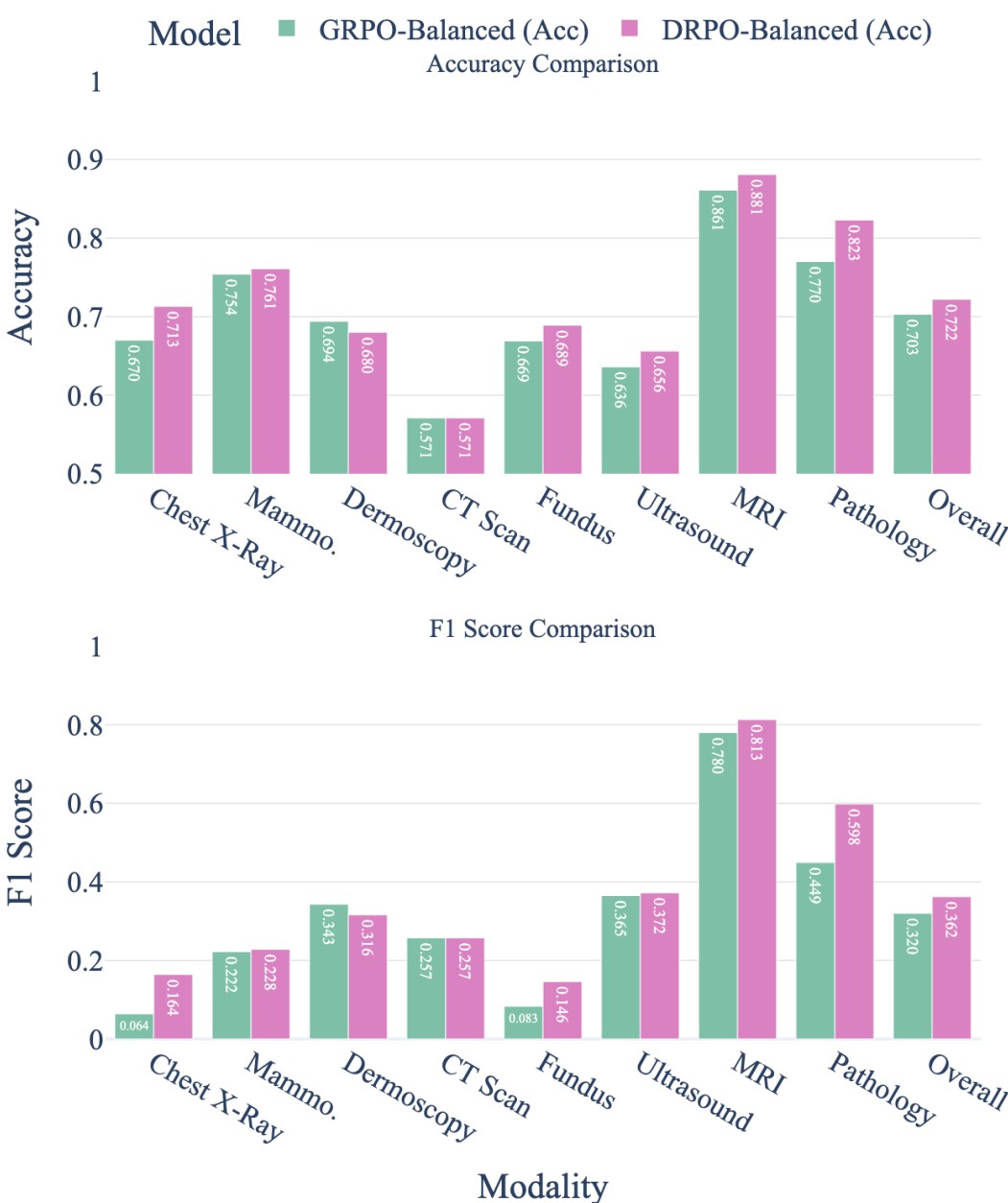

Figure 5: **Comparison of DRPO and GRPO on Balanced Datasets.** Acc: Accuracy, F1: F1 Score. To remove the influence of imbalanced dataset, we further conducted a experiment on a balanced subset of the 30 datasets, where each dataset share the same portion in the training dataset mix. This helps us compare our method with similar methods like loss scaling (i.e. focal loss[53]) and upsampling/downsampling techniques. Thanks to the dynamic weighting based on both difficulty and scarcity, our method better captures the changing dynamics throughout the training, allowing it to perform better than GRPO even with a perfectly balanced dataset.

Table 6: **Details of individual datasets used in training of QoQ-Med.**

| Dataset | # Classes | Classes |
|---|---|---|
| PTB-XL[82] | 7 | Normal, Conduction Delay (CD), Hypertrophy (HYP), Myocardial Infarction (MI), Sinus Tachycardia/Bradycardia/Conduction (STTC), Atrial Fibrillation/Atrial Flutter (A. Fib/Aflutter), Other |
| Chapman-Shaoxing[93] | 7 | Same as PTB-XL |
| Georgia[4] | 7 | Same as PTB-XL |
| CPSC[54] | 7 | Same as PTB-XL |
| CheXpert[38] | 14 | Atelectasis, Cardiomegaly, Consolidation, Edema, Enlarged Cardiomediastinum, Fracture, Lung Lesion, Lung Opacity, Pleural Effusion, Pneumonia, Pneumothorax, Pleural Other, Support Devices, No Finding |
| MIMIC-CXR[40] | 14 | Same as CheXpert |
| VinDr-CXR[59] | 6 | Lung tumor, Pneumonia, Tuberculosis, COPD, Other diseases, No finding |
| COVID-19[15] | 4 | Normal, Bacterial Pneumonia, COVID-19, Viral Pneumonia |
| CoronaHack[58] | 3 | Normal, Bacterial Pneumonia, Viral Pneumonia |
| VinDr-Mammo[68] | 5 | BI-RAD 1-5 |
| CBIS-DDSM[69] | 6 | BI-RAD 0-5 |
| CMMD[18] | 2 | Benign, Malignant |
| ISIC-2020[71] | 2 | Malignant, Benign |
| HAM10000[81] | 5 | Melanoma (MEL), Nevus (NV), Basal Cell Carcinoma (BCC), Actinic Keratosis/Intraepithelial Carcinoma (AKIEC), Other (OTHER) |
| PAD-UFES-20[64] | 5 | Melanoma (MEL), Nevus (NV), Basal Cell Carcinoma (BCC), Actinic Keratosis/Intraepithelial Carcinoma (AKIEC), Other (OTHER) |
| Messidor-2[23] | 5 | None, Mild DR, Moderate DR, Severe DR, PDR |
| APTOS 2019[8] | 5 | No DR, Mild, Moderate, Severe, Proliferative DR |
| Jichi[78] | 3 | SDR (simple diabetic retinopathy), PPDR (pre-proliferative diabetic retinopathy), PDR (proliferative diabetic retinopathy) |
| LNDb[67] | 3 | nodule $\geq$ 3mm, nodule <3mm, non-nodule |
| INSPECT[35] | 5 | No PE, Acute Subsegmental-only PE, Acute PE, Subsegmental-only PE, Chronic PE |
| KiTS23[30] | 2 | Benign, Malignant |
| Hemorrhage[32] | 2 | No Hemorrhage, Has Hemorrhage |
| RSPECT[16] | 3 | No PE, Chronic PE, Acute PE |
| BUSI[3] | 3 | Normal, Malignant, Benign |
| COVID-BLUES[84] | 2 | Has COVID, No COVID |
| COVID-US[25] | 3 | Covid, Pneumonia, Normal |
| Brain Tumor[10] | 4 | No Tumor, Pituitary Tumor, Glioma Tumor, Meningioma Tumor |
| Brain Tumor 2 | 2 | Yes, No (presence of tumors) |
| LC25000[13] | 5 | Colon adenocarcinomas, Benign colon, Lung adenocarcinomas, Lung squamous cell carcinomas, Benign lung |
| BCSS[7] | 4 | Tumor, Stroma, Lymphocytic infiltrate, Necrosis/debris |

domains. In this work, we use the vision (2D and 3D) and ECG subset of the CLIMB dataset, which contains 707K 2D, 1.83M 3D, and 78.9K ECG data.

For RQ1, we use the same training/validation split as in the original CLIMB dataset, which largely inherits the splits from the original papers.

A list of datasets used in the paper is included in Table 6.

## C.1 QA Pair Generation

In particular, we use the QA version of the datasets, created in the following way:

For every sample $x_i \in D_k$ with label set $y_i \subseteq V_k$, a paired question and answer $(q_i, a_i)$ is generated as follows:

- **Question** $q_i$: a natural-language query formed by combining
    - a brief task prompt (e.g., "identify the primary abnormality"),
    - a cue about the input modality (e.g., "on the chest X-ray"),
    - the list of allowable labels drawn from $V_k$.

- **Answer** $a_i$**:** the gold label derived from $y_i$, formatted as
    - a single class $a_i \in V_k$ when categories are mutually exclusive, or
    - a subset $a_i \subseteq V_k$ when multiple findings may co-occur.

Formally, we introduce a mapping $\psi : (x_i, y_i, V_k) \mapsto (q_i, a_i)$ that converts each classification instance into its question-answer analogue without altering the underlying task semantics. As an illustration, consider the CheXpert dataset for chest X-ray diagnosis:

> **Question.** "Shown above is a frontal chest radiograph. State the predominant finding, choosing from: No Finding, Cardiomegaly, Pleural Effusion, Lung Consolidation, Atelectasis."
>
> **Answer.** "Pleural Effusion"

For multi-label tasks, predictions are judged with an order-independent match: a response $\hat{a}_i$ is deemed correct iff $\hat{a}_i = a_i$, irrespective of label ordering. This criterion accommodates scenarios where several conditions may be present simultaneously and listed in any sequence.

### C.2  Defintion of Novel Tasks, Understudied Modalities and Underrepresented Regions

We follows the original work's definition of Novel Tasks, Understudied Modalities and Underrepresented Regions:

**Novel Tasks.**    Novel tasks are defined as new diseases or conditions not present previously. Examples include COVID.

**Understudied Modalities.**    Building on the criteria set forth in the original study, we characterize each modality along two axes: *(i) research attention*, estimated by the number of Google Scholar hits returned for standardized queries such as "[modality] classification," and *(ii) public data availability*, counted as the total number of openly released samples. Based on both criteria, mammography and ECG appear under-studied in the literature, whereas ultrasound is chiefly limited by data scarcity.

**Underrepresented Regions.**    The underrepresented regions are defined as datasets collected from regions historically underrepresented regions (like Africa, South America and Parts of South and Southeast Asia), or regions with less economic developments (like India, Vietnam, Iraq and Brazil).

## D  Details of Training and Evaluation

### D.1  Training Hyperparameters

Unless mentioned otherwise, we use the same set of hyperparameters to train the model across different training methods. The models are trained for 1 epoch on an 8xNVIDIA A100 and H200 GPU instances. For 7B models, we use a per-device batch size of 4, and a rollout batch size of 512. The maximum context length is 8192. To ensure consistency throughout the training, we shuffle the data with seed 42 beforehand, and disable shuffling throughout the training process. To save compute, we employ early stopping, which stops training when the accuracy converges and stops improving. Most 7B model trainings converge within 2 days of training. The training of 32B model takes more than 2 weeks to train on an 8xA100 machine, so a 8xH200 machine is used to speed up the training process of 32B model via faster interconnect.

We build our training pipeline based on the FSDP and VeRL framework, with vLLM to speed up reasoning training with KV Cache. We use a learning rate of 1e-6, a weight decay of 1e-2, and a KL coefficient of 1e-4. We use AdamW full model training at 32-bit precision for all 7B models, and at 16-bit precision for the training of the 32B model.

Throughout the training and evaluation, the 3D images and videos are sliced uniformly into 4 frames, before getting concatenated into the model's input. Images are downsized so that they have a max pixel count of 524,288. For all RL training methods, we use Qwen2.5-VL-7B [9] as the base model. For QvQ-Med, an ECG encoder named ECG-JEPA [43] is prepended.

**Environmental Impacts.** Experiments on 7B/32B models exhibit minimal environmental impacts. Our A100 SXM has a TDP of 400W. Assuming CPUs and other components have a combined TDP of 600W, training of a 7B model requires an electricity of 182.4kWh. Under the same assumption, training a 32B model requires an electricity of 571.2 kWh. We would also like to point out that releasing public models as we did has a positive environmental impact, as researchers can directly use or finetune the model on the domain data, instead of doing the training repeatedly.

The following sections detail the evaluation metrics used in our experiments to assess the performance of DRPO and baseline methods across various clinical tasks and datasets.

## D.2 RQ1: Performance Comparison with Critic-free RL Methods

For comparing DRPO with other critic-free RL methods and models, we formulate each clinical diagnosis task as a multi-label classification problem over all possible labels in a dataset. The evaluation metrics are computed as follows:

$$\text{Accuracy}_{\text{label}} = \frac{TP + TN}{TP + TN + FP + FN} \tag{8}$$

$$\text{Precision}_{\text{label}} = \frac{TP}{TP + FP} \tag{9}$$

$$\text{Recall}_{\text{label}} = \frac{TP}{TP + FN} \tag{10}$$

$$\text{F1}_{\text{label}} = 2 \cdot \frac{\text{Precision}_{\text{label}} \cdot \text{Recall}_{\text{label}}}{\text{Precision}_{\text{label}} + \text{Recall}_{\text{label}}} \tag{11}$$

where $TP$, $TN$, $FP$, and $FN$ represent true positives, true negatives, false positives, and false negatives, respectively, for each label across all samples.

To account for potential class imbalance, we compute the balanced accuracy for each dataset:

$$\text{Balanced Accuracy}_{\text{dataset}} = \frac{1}{|L|} \sum_{l \in L} \text{Accuracy}_{\text{label}} \tag{12}$$

where $L$ is the set of all labels in the dataset.

Similarly, we compute the macro-F1 score for each dataset:

$$\text{Macro-F1}_{\text{dataset}} = \frac{1}{|L|} \sum_{l \in L} \text{F1}_{\text{label}} \tag{13}$$

To obtain domain-level metrics, we perform unweighted averaging across all datasets within each clinical domain:

$$\text{Metric}_{\text{domain}} = \frac{1}{|D_{\text{domain}}|} \sum_{d \in D_{\text{domain}}} \text{Metric}_{\text{dataset}} \tag{14}$$

where $D_{\text{domain}}$ represents the set of datasets in a specific clinical domain, and Metric can be either balanced accuracy or macro-F1.

Finally, we compute the overall performance by averaging the domain-level metrics across all 9 clinical domains:

$$\text{Metric}_{\text{overall}} = \frac{1}{9} \sum_{i=1}^{9} \text{Metric}_{\text{domain}_i} \tag{15}$$

### D.3 RQ2: Multimodal Input Handling

For evaluating the model's ability to handle mixed multimodal inputs on the MIMIC-IV dataset, we use similar metrics as in RQ1. We formulate two prediction tasks:

#### D.3.1 Length of Stay (LOS) Prediction

LOS prediction is formulated as a 4-class classification problem. Patient stays are binned into the following categories:

- Class A: 0-4 days
- Class B: 5-8 days
- Class C: 9-12 days
- Class D: more than 12 days

#### D.3.2 48-hour In-Hospital Mortality (48-IHM) Prediction

48-IHM is formulated as a binary classification task. A positive label is assigned if either:

- The patient's death date is within 48 hours of admission, or
- The patient is discharged to hospice care within 48 hours of admission

For both tasks, we compute accuracy and F1 scores. For the multi-class LOS prediction, we compute:

$$\text{Accuracy}_{\text{LOS}} = \frac{\text{Number of correct predictions}}{\text{Total number of samples}} \tag{16}$$

$$\text{Macro-F1}_{\text{LOS}} = \frac{1}{4} \sum_{c \in \{A,B,C,D\}} \text{F1}_c \tag{17}$$

For the binary 48-IHM prediction:

$$\text{Accuracy}_{\text{48-IHM}} = \frac{TP + TN}{TP + TN + FP + FN} \tag{18}$$

$$\text{F1}_{\text{48-IHM}} = 2 \cdot \frac{\text{Precision} \cdot \text{Recall}}{\text{Precision} + \text{Recall}} \tag{19}$$

where Precision and Recall are computed as defined in RQ1.

The input data for both tasks consists of:

- Chest X-ray images from MIMIC-IV-CXR
- 12-lead ECG traces from MIMIC-IV-ECG
- Textual electronic health record data from MIMIC-IV, including vital signs, lab measurements, treatments, medications, and demographics collected within 48 hours of admission

Data without timestamps (e.g., diagnoses) and discharge summaries and radiology reports are excluded from the input.

### D.4 RQ3: Quality of Reasoning Traces and Bounding Boxes

To evaluate the quality of reasoning traces and bounding boxes produced by DRPO, we employ both quantitative and qualitative measures.

### D.4.1 Bounding Box Quality

For quantitatively assessing bounding box quality, we compute the Intersection over Union (IoU) against ground truth segmentations:

$$\text{IoU} = \frac{|B_{\text{pred}} \cap B_{\text{gt}}|}{|B_{\text{pred}} \cup B_{\text{gt}}|} \tag{20}$$

where $B_{\text{pred}}$ is the predicted bounding box and $B_{\text{gt}}$ is the ground truth segmentation.

For datasets without available ground truth segmentations, we generate segmentations using the BiomedParse model. Samples with no meaningful segmentations are excluded from this evaluation.

### D.4.2 Reasoning Trace Quality

For qualitative assessment of reasoning traces, we collaborate with clinicians to evaluate the relevance of each reasoning statement to the final diagnosis. The evaluation categorizes statements into three levels:

- **High Relevance**: The statement is critical and closely related to the final diagnosis.
- **Low Relevance**: The statement relates to relevant medical concepts but is not necessary to arrive at the diagnosis.
- **No Relevance**: The statement is not helpful to the final diagnosis process.

We compute the distribution of these relevance categories across the validation set of both CLIMB and MIMIC-IV datasets to assess the overall quality of the model's reasoning process.

# E    Relevance Annotation of Reasoning Traces

To better understand the reasoning process and help model produce better reasoning in the future, we collaborated with clinicians to annotate the reasoning traces on randomly selected samples in the validation set, which is representative of model's real world performance on unseen samples. We present unfiltered, original model reasoning traces here with color-coded highlights indicating relevance. Phrases marked in yellow represent information of low clinical relevance, pink highlights indicate high clinical relevance, and aqua denotes content judged to have no relevance to the diagnosis.

---

**Question:** Above is a brain MRI scan of a patient. What type of tumor, if any, is present in this brain MRI scan?

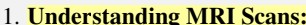

**Answer with one of the following:** No Tumor, Glioma Tumor, Meningioma Tumor, Pituitary Tumor

**Ground Truth Label:** No Tumor

**Model's Prediction:** No Tumor

**Reasoning:** To analyze the brain MRI scan, we should consider the following steps:

1. **Understanding MRI Scans:**

   - **MRI (Magnetic Resonance Imaging)** is a diagnostic tool that uses a powerful magnet, radio waves, and a computer to create detailed images of organs and structures inside the body, including the brain.
   - It is particularly useful for identifying tumors because it can show differences in tissue density, which can indicate whether a mass is solid, liquid, or blood-filled, and whether it is benign or malignant.

2. **Identify Key Features of a Tumor on MRI:**

   - **Hypointense/Isointense on T1:**
     - Most normal brain tissue appears dark on a T1-weighted MRI, so a tumor that is the same color as normal brain tissue is called "isointense" or "hypointense."
     - This means it is often harder to spot on a T1 image, but can be seen on the contrast-enhanced images.
     - A tumor that is the same color as brain tissue on a T1-weighted MRI is often a sign of a glioma, which is a type of brain tumor that originates from glial cells in the brain.
   - **Hyperintense on T2:**
     - A tumor that appears bright on a T2-weighted MRI is called "hyperintense."
     - This means it is often easier to see on a T2 image, but is a less specific sign.
     - It can be seen in many brain tumors, including gliomas, meningiomas, and pituitary tumors.
     - The brightness on T2 images is often associated with edema, which is a sign of inflammation and swelling around the tumor, which is a common feature of many types of tumors.
     - The image provided is a T2 weighted MRI, which usually makes tumors appear bright, and in this case, the tumor appears hyperintense, consistent with a typical appearance of a meningioma on MRI. Meningiomas are often hyperintense on T2 images and hypointense on T1 images, which is what we see here. They are usually benign but can be quite large and cause symptoms. However, without further context, it is impossible to determine definitively the type of tumor. The MRI is simply a tool, and a definitive diagnosis often requires a combination of imaging, a patient's medical history, and other tests such as a biopsy.

---

**Question:** Above is a chest X-ray image of a patient. What is the diagnosis of the patient in the X-ray image?

**Answer with one or multiple phrases from the following:** No Finding, Enlarged Cardiomediastinum, Cardiomegaly, Lung Opacity, Lung Lesion, Edema, Consolidation, Pneumonia, Atelectasis, Pneumothorax, Pleural Effusion, Pleural Other, Fracture, Support Devices

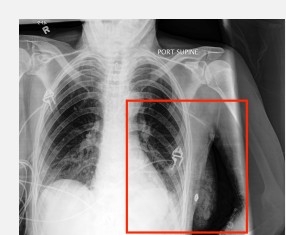

**Ground Truth Label:** Support Devices

**Model's Prediction:** No Findings

**Reasoning to annotate:** To analyze the chest x-ray image, we need to consider the following steps:

1. **Examine the Image for Anomalies:**

- **Size and Shape of the Heart and Mediastinum:** The mediastinum is the area in the chest that contains the heart, great vessels, trachea, and esophagus. In this chest x-ray, the mediastinal area appears to be enlarged, which is a strong indicator of pathologies such as cardiomegaly (an enlarged heart) or other mediastinal pathologies, but it's essential to understand that this is an interpretation and should be confirmed by a medical professional.
- **Lung Opacity:** The lung fields are evaluated for any signs of increased density, which can indicate fluid, air, or tissue abnormalities. The x-ray shows increased density in the lung fields, which can suggest consolidation, such as from pneumonia, or other pathology. The appearance of the lung fields is consistent with signs of fluid, which is a sign of pleural effusion.
- **Support Devices:** The x-ray shows a pacemaker, which is a medical device used to regulate the heartbeat. This is a normal finding in some patients and does not indicate pathology.

2. **Consider the X-ray Protocol and Patient Information:**

- The patient's position (supine) is noted. This position is often used in chest x-rays to ensure that the patient is in a comfortable position and to ensure that the image is clear and useful for interpretation. The radiographic markers (e.g., the "P" and "R" signs) are standard for x-rays to indicate the position of the patient and to ensure that the image is correctly oriented.

3. **Analysis of the Image for Any Indications of Pathology:**

- **Pleural Effusion:** The increased density in the lung fields is indicative of pleural effusion, which is a buildup of fluid in the pleural space, the area between the lungs and the chest wall. This is a common sign of heart failure, liver disease, or malignancy.
- **Cardiomegaly:** The enlarged heart size, though not definitive without further clinical context, is suggested by the mediastinal widening, though the definitive interpretation is that this is more about the mediastinum and not a direct indicator of cardiomegaly.

Based on the analysis, the most likely findings in this x-ray image are suggestive of:

- Pleural Effusion due to the increased density seen in the lung fields.
- No Finding for other abnormalities, given that there are no signs of fractures, pneumothorax, or pneumonia.

**Question:** Above is a lung ultrasound video. What is the diagnosis based on this lung ultrasound?

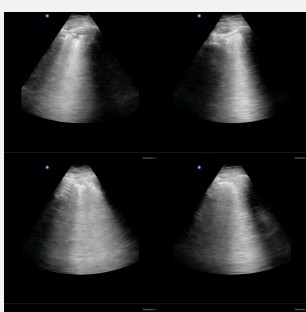

**Answer with one word from the following options:** covid, pneumonia, normal

**Ground Truth Label:** covid

**Model's Prediction:** pneumonia

**Reasoning to annotate:** To analyze the image and determine the reasoning process for making a diagnosis, we follow these steps:

1. **Examine the Image Type and Purpose**: The image is a lung ultrasound, which is a diagnostic tool used to assess lung parenchyma (the part of the lung that contains air and is responsible for gas exchange) for signs of abnormalities. Ultrasound is particularly useful in the early stages of respiratory disease, as it can detect changes in lung tissue that may be indicative of various conditions.

2. **Understanding Ultrasound Features**: In ultrasound, different tissue types and pathologies can have distinct appearances. For example:

   - **Hyperechoic (bright) areas** often indicate a pathological process, such as air, fluid, or solid tissue. In lung ultrasound, hyperechoic areas might suggest consolidation, which is often associated with pneumonia.
   - **Hypoechoic (dark) areas** are usually indicative of normal lung tissue, but in some cases, they could also be associated with certain pathologies.
   - **Echogenicity patterns** can vary in different lung pathologies, and these patterns can help in identifying the nature of lung pathology. For example, the image shows a relatively uniform, hyperechoic pattern, which is often seen in lung ultrasound when it is used to identify areas of consolidation, a hallmark of pneumonia.

3. **Clinical Correlation**: It is essential to consider the patient's clinical context, such as symptoms, medical history, and other diagnostic tools used in conjunction with the ultrasound. However, since the image is just an ultrasound, we must rely on the image's characteristics alone. The image shows a single ultrasound frame, and often, a series of ultrasound images are analyzed over time to get a comprehensive view of lung involvement, but for this exercise, we are only analyzing one image.

4. **Interpretation of Image**: In this case, the image shows a pattern that is consistent with lung ultrasound findings that suggest pneumonia. The hyperechoic areas are characteristic of consolidation, which is a sign of fluid or air accumulation in the lungs, typically associated with pneumonia. This is further supported by the fact that lung ultrasound is a sensitive tool for identifying pneumonia in its early stages, often before radiographic changes are visible.

**Question:** Above is a lung ultrasound video. What is the diagnosis based on this lung ultrasound?

**Answer with one word from the following options:** covid, pneumonia, normal

**Ground Truth Label:** covid

**Model's Prediction:** pneumonia

**Reasoning to annotate:** To analyze the image and determine the reasoning process for understanding what the image might suggest, follow these steps:

1. **Examine the Image Type:** The image is a lung ultrasound, which is a diagnostic imaging technique used in medicine to visualize the lung parenchyma. Ultrasound is often used in emergency settings for quick, non-invasive assessment, especially in cases where quick decision-making is needed. Ultrasound images are grayscale, with varying shades of gray representing different tissue densities. The image shown here is a grayscale ultrasound image, which is typical for ultrasound.

2. **Analyze the Image Features:**

  - **Grayscale Intensity:** The image is grayscale, with varying shades of gray indicating different tissue densities. In ultrasound, the brightness of the image can be indicative of different pathologies. For example:

  - **Hyperechoic (brighter than normal tissue):** This often indicates air, which is why ultrasound is used to visualize the chest, as air is easily detected. In this image, the lesion appears hyperechoic, meaning it reflects more ultrasound waves than the surrounding tissue, which is a characteristic sign that can suggest the presence of fluid or air, which is often associated with disease such as pneumonia.
  - **Hypoechoic (darker than normal tissue):** This indicates less reflective tissue and is often associated with soft tissues, such as tumors, or areas of fluid accumulation. The lesion in this image is not hyperechoic, so it is unlikely to suggest air, but rather, it suggests a lesion that is darker than the surrounding lung tissue, which could be indicative of a pathological condition.
  - **Shape, Size, and Border:** The lesion in the image is well-defined, meaning it has a clear border, which is a feature that can be associated with malignancy or inflammation. The shape is mostly round, which is common in benign lesions. However, the exact nature of the lesion would need further investigation to determine if it is benign or malignant. The size and location are critical, but these cannot be determined from a single image.

3. **Clinical Context:** Ultrasound is often used in the context of acute respiratory distress, and it is frequently used to identify signs of pneumonia, which can be associated with various signs on ultrasound such as consolidation, pleural effusion, or air. The image shows a lesion, which may suggest pathology, but it is crucial to understand the clinical context, such as the patient's history, symptoms, and other clinical signs, to make a full diagnosis.

Based on the reasoning process above, the analysis of this ultrasound image suggests that it is showing signs that might be indicative of pneumonia. The hyperechoic nature of the lesion is often associated with consolidation, a sign that can occur in pneumonia. However, it's important to remember that ultrasound is a tool and can only suggest possibilities; the final diagnosis is typically confirmed by further tests, such as chest X-rays, CT scans, or other medical procedures.

**Question:** Above is a chest X-ray image of a patient. What is the diagnosis of the patient in the X-ray image?

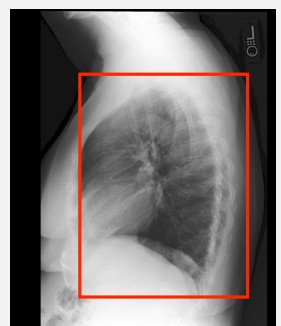

**Answer with one or multiple phrases from the following:** Atelectasis, Cardiomegaly, Consolidation, Edema, Enlarged Cardiomediastinum, Fracture, Lung Lesion, Lung Opacity, No Finding, Pleural Effusion, Pleural Other, Pneumonia, Pneumothorax, Support Devices

**Ground Truth Label:** Pneumonia

**Model's Prediction:** Pneumonia

**Reasoning to annotate:** To analyze the chest x-ray image, we need to understand the reasoning process for identifying signs that may suggest a particular diagnosis. The reasoning process involves observing the following key features:

2. **Lung Opacity:**

- The density of the lung fields is evaluated. Normally, lung tissue is very aerated and appears very light on the x-ray, almost completely black. If there is any increased opacity in the lung fields, it may indicate fluid, blood, or abnormal tissue, such as consolidation, atelectasis, or consolidation, which are suggestive of pneumonia. In this image, the lung fields appear darker than normal, which is a sign of increased opacity. This could indicate a number of pathologies, such as consolidation, which is often associated with pneumonia. Atelectasis, which is the collapse of part of the lung, can also appear as a darker opacity on x-ray and is a common cause of increased lung opacity.

3. **Pleural Effusion:**

- This is where there is an abnormal accumulation of fluid in the pleural space, which can be seen on x-rays as a smooth, curved opacity at the base of the lung, often associated with lung collapse.
- This isn't visible in this image, as there's no indication of a smooth, curved opacity.

4. **Pneumothorax:**

- This is where air is in the pleural space, and it is usually seen as a black, triangular area on x-rays, because air is very radiolucent and doesn't allow any x-rays to pass through.
- This image does not have any signs of pneumothorax, such as a triangular black area.

5. **Pleural Other:**

- This can include other pathologies that affect the pleura, such as malignancy, infection, or inflammation. However, this image does not show signs of malignancy or inflammation, so it is unlikely to be the primary focus.

6. **No Finding:**

- This means that the x-ray shows no obvious signs of pathology, but this is unlikely in this image, as there are signs that suggest pathology.

7. **Lung Lesion:**

- This is a general term for any abnormality in the lung, which can be a sign of pathology. The darker, more opaque areas in the lungs are suggestive of a lung lesion, which could be caused by a variety of reasons including pneumonia, atelectasis, or consolidation. In this image, the areas of increased opacity in the lung fields are suggestive of a lung lesion.

Based on the reasoning process, the most plausible findings in this image are those that suggest increased lung opacity, which are suggestive of pathology. The reasoning process points towards lung lesions, which are often associated with pneumonia, atelectasis, or consolidation. The signs of increased lung opacity are indicative of a pathology that could be pneumonia, which is a common cause of increased lung opacity in x-rays.

**Question:** Above is a brain MRI scan. Is there a tumor present in this brain MRI scan?

**Answer with one of the following:** No Tumor, Has Tumor

**Ground Truth Label:** Has Tumor

**Model's Prediction:** Has Tumor

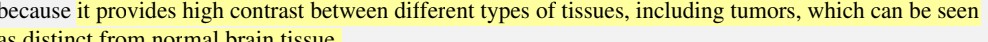

**Reasoning to annotate:** To analyze the image, let's use the following reasoning steps:

1. **Identify the Type of Image:** The image is a brain MRI scan, which is a medical imaging technique used to visualize detailed cross-sectional images of the brain. MRI is particularly useful for detecting tumors because it provides high contrast between different types of tissues, including tumors, which can be seen as distinct from normal brain tissue.

2. **Analyze the Image Features:**

- **Shape and Size:** The tumor appears as a round, well-circumscribed lesion, meaning it has a clear boundary from the surrounding brain tissue. In MRI images, tumors are often characterized as having a distinct, often irregular, boundary, while normal brain tissue has a smooth, well-defined edge. In this case, the tumor is clearly distinct from the surrounding brain tissue, which is a strong indicator of a lesion that could be a tumor.
- **Signal Intensity:** The signal intensity refers to how bright or dark the tissue appears on the MRI image. Tumors can vary in signal intensity, but often, they appear brighter than normal brain tissue on a T2-weighted MRI (which is the standard MRI sequence used for brain imaging). This is because tumors have different tissue properties, such as increased blood flow, which can lead to a higher signal intensity on MRI. This tumor is indeed bright on this image, suggesting it is a high-intensity lesion, which is often associated with tumors, but this alone is not definitive for a diagnosis.
- **Location and Location:** The tumor is located in the brain, and in this case, it's in a region where tumors are common. The location and size of the lesion are important factors in determining the potential pathology, but they are not definitive on their own. The location in this image is consistent with where tumors commonly appear in the brain, and the size is indicative of a lesion that could be a tumor.
- **Clinical Context:** While this reasoning relies mostly on the image itself, the context (clinical history of the patient, symptoms, and any other imaging studies) would be considered by a medical professional to make a definitive diagnosis. This context is often essential for a precise understanding and interpretation of the image.

**Question:** Above is a brain CT scan slice. Is there any hemorrhage present in this CT scan slice?

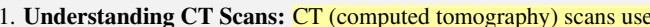

**Answer with one of the following:** No Hemorrhage, Has Hemorrhage

**Ground Truth Label:** No Hemorrhage

**Model's Prediction:** No Hemorrhage

**Reasoning to annotate:** To analyze the image, we will go through a series of steps to understand what we're looking at and what it might tell us about the presence of hemorrhage.

1. **Understanding CT Scans:** CT (computed tomography) scans use X-rays and computer processing to create detailed images of the body's internal structures. They are particularly useful for diagnosing brain injuries because they can show changes in tissue density, which can indicate bleeding, tumors, or other pathologies. The color and density of the tissue on a CT scan can often give clues about what is happening.

2. **Anatomy of a CT Scan Image:** A CT scan is a cross-sectional view of the body, and each slice provides a detailed look at the brain at that level. The image is typically grayscale, with different shades representing different densities. The brain tissue usually appears as a medium gray, while air or gas appears black, and bone, which is very dense, appears white.

3. **Color Coding on CT Scans:** The way tissues appear on CT scans can vary depending on the type of contrast used and the type of scan (e.g., CT with contrast, CT without contrast). In a standard CT scan, brain tissue usually appears as a medium gray. If there is bleeding, the blood can appear in different shades on CT scans, depending on the type and age of the hemorrhage. Typically, blood is denser than brain tissue and appears white on a non-contrast CT scan, while on a contrast-enhanced CT scan, it can appear darker (hypodense) or have a different color depending on its age and location.

4. **Symptoms of Hemorrhage on CT:** On a CT scan, signs of hemorrhage include:

   - **Hypodense (darker than normal brain tissue) on non-contrast CT**: This is the most common appearance for acute hemorrhage. It happens when blood clots and does not allow contrast to pass through.
   - **Hyperdense (whiter than normal brain tissue) on non-contrast CT**: This is seen with older blood, where the blood has already clotted and is more dense than normal brain tissue.
   - **Contrast enhancement**: If a contrast medium is used, it can help highlight the blood clot, making it stand out even more on the scan.

5. **Analyzing the Image:**

   - The image is in grayscale, and in this case, the brain tissue appears in a medium gray shade, which is normal for brain tissue in a CT scan.
   - There is no evidence of white areas (hyperdense), which would suggest older blood clots or contrast enhancement that would indicate bleeding. The entire brain tissue appears uniformly gray, without any obvious bright or dark areas that would suggest hemorrhage.

Based on these considerations, we are looking for signs that would indicate hemorrhage, but there are none in this image. The tissue density is consistent with normal brain tissue, and there is no evidence of blood clots or other signs of hemorrhage. The image is consistent with a brain CT without hemorrhage.

**Question:** Above is a dermoscopy image of a patient. What is the diagnosis of the patient in the dermoscopy image?

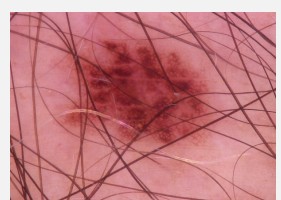

**Answer with one word from the following:** Melanoma, Nevus, Basal Cell Carcinoma, AKIEC, Other

**Ground Truth Label:** Nevus

**Model's Prediction:** Nevus

**Reasoning to annotate:** To analyze the image, let's go through the reasoning process step by step:

1. **Examine the Image:**

   - The image shows a close-up of skin with a lesion. The lesion appears to have a somewhat uneven, irregular, and slightly raised surface, which can be indicative of various skin conditions. The coloration is mostly a dark brown, which could be a sign of a pigment-rich lesion, such as a mole or a nevus. The border of the lesion appears somewhat uneven, which is a characteristic feature of many nevi, but this is not definitive without further context.

2. **Consider the Clinical Context:**

   - In dermatology, dermoscopy is often used to evaluate skin lesions for signs that may suggest malignancy. The image provided does not have any clinical history or patient data, but based on the visual clues, we can consider the following points:
     - **Melanoma** typically shows a variety of features that can be indicative of malignancy, such as a changing color, irregular border, or asymmetry. The lesion in this image does not show any obvious signs of malignancy like ulceration, bleeding, or bleeding into the lesion, which are often associated with melanoma. However, these signs are not always present, and a definitive diagnosis requires a biopsy.
     - **Nevus** (or mole) is a common benign growth of the skin. These are often characterized by their uniformity, color, and border. In this image, the lesion's color is consistent, and its border is somewhat irregular, but this is not a definitive sign for malignancy. The lesion could be a nevus, but we cannot make a certain diagnosis without further clinical context and examination.
     - **Basal Cell Carcinoma** (BCC) is usually a non-melanoma skin cancer. BCCs are often erythematous, ulcerated, and can be very vascular. The lesion in this image lacks these characteristics, so it is less likely to be BCC. However, it is still a possibility, but it is less likely based on the image alone.
     - **AKIEC** (Actinic Keratoses) are precancerous skin lesions that are usually caused by chronic sun exposure. They often have a rough, scaly, or warty appearance, and they are usually a light tan, brown, or red. This lesion does not have the characteristics of AKIEC, and the image does not suggest a precancerous lesion.
     - **Other**: There are other possibilities such as a vascular lesion, which could be due to various causes such as angiomas, which are benign. However, the image does not provide enough information to make a definitive diagnosis, and further clinical examination and histopathology are required.

**Question:** Above is a clinical image of a patient. What is the diagnosis of the patient in the clinical image?

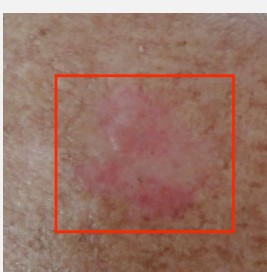

**Answer with one word from the following:** Melanoma, Nevus, Basal Cell Carcinoma, AKIEC, Other

**Ground Truth Label:** Basal Cell Carcinoma

**Model's Prediction:** Basal Cell Carcinoma

**Reasoning to annotate:** To analyze the image, let's go through the reasoning process step by step:

1. **Observation of the Image:**

   - **Color and Texture:** The skin lesion in the image is pinkish in color, which can be indicative of various skin conditions. The color suggests it's not a dark mole (nevus) or a dark, raised, and unevenly shaped mass (melanoma). The color is relatively uniform, but the pink hue is a bit unusual for normal skin, indicating something might be amiss.
   - **Shape and Size:** The lesion appears as a slightly raised, pinkish, and slightly elevated area, with a somewhat indistinct border. The uneven border is a critical feature that could suggest a basal cell carcinoma, as these often have a more irregular border, while melanoma is known for having a jagged, uneven, or indistinct border. Nevus usually have a more sharply defined border and are often flat, while AKIEC (Actinic Keratosis) lesions have a rough, scaly surface, which is more consistent with the texture in this image. The lesion's size and shape are also consistent with basal cell carcinoma, which is more likely to present as a raised, pink, or flesh-colored lesion on the skin, especially on sun-exposed areas.
   - **Skin Lesion Location and Context:** The lesion is on the skin, which is a common location for basal cell carcinomas, which are the most common type of skin cancer. They often appear on areas most exposed to the sun, such as the face, ears, neck, and back, which aligns with the location of the lesion in this image.
   - **Clinical Context:** Basal cell carcinomas are typically painless and are often associated with chronic sun exposure. They are usually benign, but they can be removed to prevent potential malignancy. In this context, the lesion's color, shape, and location are consistent with basal cell carcinoma. It is important to remember that only a dermatologist can make a definitive diagnosis, but this analysis is based on common clinical patterns.

# F   Full Training Metrics

We provide full training metrics across individual datasets for RQ1 in the following tables:

Table 7: **Performance comparison with standard deviations.** Results averaged over 4 runs. Acc: Accuracy, F1: F1 Score, CXR: Chest X-ray.

| Model | CXR | | Mammo. | | Dermoscopy | | CT Scan | |
|---|---|---|---|---|---|---|---|---|
| | **Acc** | **F1** | **Acc** | **F1** | **Acc** | **F1** | **Acc** | **F1** |
| **SFT** | .688±.030 | .078±.013 | .481±.015 | .056±.004 | .640±.029 | .158±.000 | .525±.000 | .236±.000 |
| **ReMax** | .636±.049 | .120±.030 | .577±.063 | .033±.020 | .644±.009 | .257±.018 | .567±.026 | .228±.001 |
| **RE++** | .730±.027 | .082±.032 | .660±.087 | .076±.011 | .635±.008 | .237±.079 | .529±.013 | .247±.039 |
| **RLOO** | **.752±.015** | .086±.003 | .471±.002 | .068±.006 | .636±.008 | .216±.028 | .534±.014 | .224±.042 |
| **GRPO** | .703±.012 | .095±.007 | .466±.000 | .059±.000 | .646±.003 | .244±.016 | .524±.002 | .236±.001 |
| **DRPO**DomainOnly | .693±.001 | .086±.001 | .751±.008 | .213±.005 | .679±.016 | .251±.042 | .571±.000 | .257±.000 |
| **DRPO** | .687±.005 | **.115±.014** | **.756±.008** | **.253±.006** | **.715±.006** | **.407±.006** | **.570±.012** | **.309±.013** |

| Model | Fundus | | Ultrasound | | MRI | | Pathology | | Overall | |
|---|---|---|---|---|---|---|---|---|---|---|
| | **Acc** | **F1** | **Acc** | **F1** | **Acc** | **F1** | **Acc** | **F1** | **Acc** | **F1** |
| **SFT** | **.715±.051** | .066±.010 | .548±.028 | **.235±.000** | .567±.038 | .197±.000 | .652±.000 | .083±.000 | .602±.014 | .139±.001 |
| **ReMax** | .678±.015 | .089±.006 | .547±.044 | .147±.008 | .547±.012 | .264±.018 | .706±.007 | .270±.011 | .596±.002 | .176±.001 |
| **RE++** | .672±.003 | .098±.013 | .519±.010 | .136±.012 | .651±.022 | .420±.025 | .668±.024 | .254±.026 | .621±.020 | .202±.053 |
| **RLOO** | .670±.003 | **.099±.013** | .519±.012 | .144±.014 | .658±.032 | .432±.020 | .699±.011 | .216±.024 | .611±.002 | .189±.003 |
| **GRPO** | .670±.001 | .086±.002 | .520±.009 | .146±.011 | .631±.027 | .395±.030 | .715±.004 | .286±.010 | .609±.005 | .193±.003 |
| **DRPO**DomainOnly | .669±.000 | .083±.000 | .480±.040 | .098±.031 | .733±.012 | .475±.012 | **.762±.009** | **.388±.028** | .668±.004 | .237±.004 |
| **DRPO** | .672±.004 | .093±.008 | **.555±.014** | .223±.021 | **.789±.019** | **.625±.028** | .708±.010 | .265±.016 | **.666±.009** | **.295±.013** |

Table 8: **Performance metrics of QoQ-Med-7B trained with DRPO$_{DomainOnly}$ (DRPO without intra-domain scaling) on Vision Only Data**

| Dataset | Accuracy | Sensitivity | Specificity | F1 | Precision | Recall |
|---|---|---|---|---|---|---|
| **Individual Datasets** | | | | | | |
| CT/INSPECT | 0.6835 | 0.2000 | 0.8000 | 0.0691 | 0.0417 | 0.2000 |
| CT/HEMORRHAGE | 0.5000 | 0.5000 | 0.5000 | 0.3333 | 0.2500 | 0.5000 |
| CT/KITS23 | 0.6154 | 0.5000 | 0.5000 | 0.3810 | 0.3077 | 0.5000 |
| CT/LNDB | 0.5000 | 0.5000 | 0.5000 | 0.3333 | 0.2500 | 0.5000 |
| CT/RSPECT | 0.5556 | 0.3333 | 0.6667 | 0.1667 | 0.1111 | 0.3333 |
| CHEST_XRAY/VINDR | 0.7465 | 0.0701 | 0.9313 | 0.0075 | 0.0241 | 0.0701 |
| CHEST_XRAY/MIMIC-CXR | 0.7694 | 0.0950 | 0.9402 | 0.0516 | 0.0606 | 0.0950 |
| CHEST_XRAY/CHEXPERT_FULL | 0.7523 | 0.1116 | 0.9342 | 0.0873 | 0.0722 | 0.1116 |
| CHEST_XRAY/CORONAHACK | 0.5556 | 0.3333 | 0.6667 | 0.1667 | 0.1111 | 0.3333 |
| CHEST_XRAY/COVID19 | 0.6481 | 0.2500 | 0.7500 | 0.1143 | 0.0741 | 0.2500 |
| FUNDUS/JINCHI | 0.6250 | 0.2500 | 0.7500 | 0.1000 | 0.0625 | 0.2500 |
| FUNDUS/APTOS | 0.6800 | 0.2000 | 0.8000 | 0.0671 | 0.0403 | 0.2000 |
| FUNDUS/MESSIDOR-2 | 0.7021 | 0.2000 | 0.8000 | 0.0814 | 0.0511 | 0.2000 |
| DERM/HAM10000 | 0.7067 | 0.2667 | 0.8167 | 0.1536 | 0.1407 | 0.2667 |
| DERM/ISIC2020 | 0.5833 | 0.5833 | 0.5833 | 0.4958 | 0.7727 | 0.5833 |
| DERM/PAD_UFES_20 | 0.7434 | 0.3167 | 0.8341 | 0.2094 | 0.2417 | 0.3167 |
| MAMMO/CBIS | 0.6694 | 0.2333 | 0.8096 | 0.0935 | 0.0670 | 0.2333 |
| MAMMO/VINDR | 0.6800 | 0.2000 | 0.8000 | 0.0686 | 0.0414 | 0.2000 |
| MAMMO/CMMD | 0.8889 | 0.5000 | 0.5000 | 0.4706 | 0.4444 | 0.5000 |
| ULTRASOUND/BUSI | 0.6667 | 0.5000 | 0.7500 | 0.4455 | 0.5965 | 0.5000 |
| ULTRASOUND/COVID-BLUES | 0.5000 | 0.0000 | 1.0000 | 0.0000 | 0.0000 | 0.0000 |
| ULTRASOUND/COVID_US | 0.4667 | 0.3667 | 0.6825 | 0.1558 | 0.3889 | 0.3667 |
| MRI/BRAIN_TUMOR | 0.6354 | 0.2708 | 0.7569 | 0.1396 | 0.2312 | 0.2708 |
| MRI/BRAIN_TUMOR_2 | 0.8780 | 0.8615 | 0.8615 | 0.8703 | 0.8902 | 0.8615 |
| PATHOLOGY/BCSS | 0.6250 | 0.2500 | 0.7500 | 0.1000 | 0.0625 | 0.2500 |
| PATHOLOGY/LC25000 | 0.9200 | 0.8000 | 0.9500 | 0.7977 | 0.8054 | 0.8000 |
| **Modality Summaries** | | | | | | |
| CT | 0.5709 | 0.4067 | 0.5933 | 0.2567 | 0.1921 | 0.4067 |
| CHEST_XRAY | 0.6944 | 0.1720 | 0.8445 | 0.0855 | 0.0684 | 0.1720 |
| FUNDUS | 0.6690 | 0.2167 | 0.7833 | 0.0828 | 0.0513 | 0.2167 |
| DERM | 0.6778 | 0.3889 | 0.7447 | 0.2863 | 0.3850 | 0.3889 |
| MAMMO | 0.7461 | 0.3111 | 0.7032 | 0.2109 | 0.1843 | 0.3111 |
| ULTRASOUND | 0.5444 | 0.2889 | 0.8108 | 0.2004 | 0.3285 | 0.2889 |
| MRI | 0.7567 | 0.5662 | 0.8092 | 0.5050 | 0.5607 | 0.5662 |
| PATHOLOGY | 0.7725 | 0.5250 | 0.8500 | 0.4488 | 0.4340 | 0.5250 |
| **Overall Results** | | | | | | |
| Overall | 0.6790 | 0.3594 | 0.7674 | 0.2595 | 0.2755 | 0.3594 |

Table 9: **Performance metrics of QoQ-Med-32B trained with DRPO on Vision Datasets**

| Dataset | Accuracy | Sensitivity | Specificity | F1 | Precision | Recall |
|---|---|---|---|---|---|---|
| **Individual Datasets** | | | | | | |
| CT/INSPECT | 0.6835 | 0.2000 | 0.8000 | 0.0691 | 0.0417 | 0.2000 |
| CT/HEMORRHAGE | 0.5000 | 0.5000 | 0.5000 | 0.3333 | 0.2500 | 0.5000 |
| CT/KITS23 | 0.6154 | 0.5000 | 0.5000 | 0.3810 | 0.3077 | 0.5000 |
| CT/LNDB | 0.5000 | 0.5000 | 0.5000 | 0.3333 | 0.2500 | 0.5000 |
| CT/RSPECT | 0.5556 | 0.3333 | 0.6667 | 0.1667 | 0.1111 | 0.3333 |
| CHEST_XRAY/VINDR | 0.7407 | 0.1048 | 0.9168 | 0.0411 | 0.1507 | 0.1048 |
| CHEST_XRAY/MIMIC-CXR | 0.7382 | 0.1087 | 0.9028 | 0.0825 | 0.1143 | 0.1087 |
| CHEST_XRAY/CHEXPERT_FULL | 0.7096 | 0.1680 | 0.9283 | 0.0766 | 0.1213 | 0.1680 |
| CHEST_XRAY/CORONAHACK | 0.7130 | 0.5694 | 0.7847 | 0.4745 | 0.4474 | 0.5694 |
| CHEST_XRAY/COVID19 | 0.7593 | 0.4375 | 0.8289 | 0.3383 | 0.2866 | 0.4375 |
| FUNDUS/JINCHI | 0.6250 | 0.2500 | 0.7500 | 0.1132 | 0.0732 | 0.2500 |
| FUNDUS/APTOS | 0.7417 | 0.3500 | 0.8396 | 0.2269 | 0.3389 | 0.3500 |
| FUNDUS/MESSIDOR-2 | 0.7064 | 0.2083 | 0.8029 | 0.1031 | 0.0754 | 0.2083 |
| DERM/HAM10000 | 0.7400 | 0.3500 | 0.8375 | 0.2725 | 0.4761 | 0.3500 |
| DERM/ISIC2020 | 0.5000 | 0.5000 | 0.5000 | 0.3333 | 0.2500 | 0.5000 |
| DERM/PAD_UFES_20 | 0.7434 | 0.3167 | 0.8341 | 0.2628 | 0.2397 | 0.3167 |
| MAMMO/CBIS | 0.7020 | 0.2083 | 0.8052 | 0.1108 | 0.2505 | 0.2083 |
| MAMMO/VINDR | 0.6833 | 0.2000 | 0.8042 | 0.0904 | 0.0686 | 0.2000 |
| MAMMO/CMMD | 0.8889 | 0.5000 | 0.5000 | 0.4706 | 0.4444 | 0.5000 |
| ULTRASOUND/BUSI | 0.6204 | 0.4306 | 0.7153 | 0.3328 | 0.3681 | 0.4306 |
| ULTRASOUND/COVID-BLUES | 0.5000 | 0.0000 | 1.0000 | 0.0000 | 0.0000 | 0.0000 |
| ULTRASOUND/COVID_US | 0.7067 | 0.6364 | 0.8095 | 0.4467 | 0.3803 | 0.6364 |
| MRI/BRAIN_TUMOR | 0.8333 | 0.6667 | 0.8889 | 0.6457 | 0.7094 | 0.6667 |
| MRI/BRAIN_TUMOR_2 | 0.9756 | 0.9792 | 0.9792 | 0.9751 | 0.9722 | 0.9792 |
| PATHOLOGY/BCSS | 0.6510 | 0.3021 | 0.7674 | 0.1967 | 0.1501 | 0.3021 |
| PATHOLOGY/LC25000 | 0.8017 | 0.4833 | 0.8813 | 0.4754 | 0.5001 | 0.4833 |
| **Modality Summaries** | | | | | | |
| CT | 0.5709 | 0.4067 | 0.5933 | 0.2567 | 0.1921 | 0.4067 |
| CHEST_XRAY | 0.7321 | 0.2777 | 0.8723 | 0.2026 | 0.2241 | 0.2777 |
| FUNDUS | 0.6910 | 0.2694 | 0.7975 | 0.1477 | 0.1625 | 0.2694 |
| DERM | 0.6611 | 0.3889 | 0.7239 | 0.2896 | 0.3219 | 0.3889 |
| MAMMO | 0.7581 | 0.3028 | 0.7031 | 0.2239 | 0.2545 | 0.3028 |
| ULTRASOUND | 0.6090 | 0.3556 | 0.8416 | 0.2598 | 0.2495 | 0.3556 |
| MRI | 0.9045 | 0.8229 | 0.9340 | 0.8104 | 0.8408 | 0.8229 |
| PATHOLOGY | 0.7264 | 0.3927 | 0.8243 | 0.3360 | 0.3251 | 0.3927 |
| **Overall Results** | | | | | | |
| Overall | 0.7066 | 0.4021 | 0.7863 | 0.3158 | 0.3213 | 0.4021 |

Table 10: **Performance metrics of 7B model trained with GRPO on Vision Only Data**

| Dataset | Accuracy | Sensitivity | Specificity | F1 | Precision | Recall |
|---|---|---|---|---|---|---|
| **Individual Datasets** | | | | | | |
| CT/INSPECT | 0.6835 | 0.2000 | 0.8000 | 0.0691 | 0.0417 | 0.2000 |
| CT/HEMORRHAGE | 0.5000 | 0.5000 | 0.5000 | 0.3333 | 0.2500 | 0.5000 |
| CT/KITS23 | 0.3846 | 0.5000 | 0.5000 | 0.2778 | 0.1923 | 0.5000 |
| CT/LNDB | 0.5000 | 0.5000 | 0.5000 | 0.3333 | 0.2500 | 0.5000 |
| CT/RSPECT | 0.5556 | 0.3333 | 0.6667 | 0.1667 | 0.1111 | 0.3333 |
| CHEST_XRAY/VINDR | 0.7432 | 0.0947 | 0.9216 | 0.0241 | 0.0224 | 0.0947 |
| CHEST_XRAY/MIMIC-CXR | 0.7771 | 0.1042 | 0.9428 | 0.0679 | 0.0805 | 0.1042 |
| CHEST_XRAY/CHEXPERT_FULL | 0.7583 | 0.1128 | 0.9342 | 0.1061 | 0.1209 | 0.1128 |
| CHEST_XRAY/CORONAHACK | 0.5556 | 0.3333 | 0.6667 | 0.1667 | 0.1111 | 0.3333 |
| CHEST_XRAY/COVID19 | 0.6481 | 0.2500 | 0.7500 | 0.1176 | 0.0769 | 0.2500 |
| FUNDUS/JINCHI | 0.6250 | 0.2500 | 0.7500 | 0.1000 | 0.0625 | 0.2500 |
| FUNDUS/APTOS | 0.6800 | 0.2000 | 0.8000 | 0.0667 | 0.0400 | 0.2000 |
| FUNDUS/MESSIDOR-2 | 0.7021 | 0.2000 | 0.8000 | 0.0814 | 0.0511 | 0.2000 |
| DERM/HAM10000 | 0.7200 | 0.2833 | 0.8292 | 0.1998 | 0.2880 | 0.2833 |
| DERM/ISIC2020 | 0.4688 | 0.4375 | 0.5000 | 0.3043 | 0.2333 | 0.4375 |
| DERM/PAD_UFES_20 | 0.7321 | 0.2917 | 0.8268 | 0.2298 | 0.2043 | 0.2917 |
| MAMMO/CBIS | 0.6082 | 0.2000 | 0.8000 | 0.0080 | 0.0041 | 0.2000 |
| MAMMO/VINDR | 0.6800 | 0.2000 | 0.8000 | 0.0667 | 0.0400 | 0.2000 |
| MAMMO/CMMD | 0.1111 | 0.5000 | 0.5000 | 0.1000 | 0.0556 | 0.5000 |
| ULTRASOUND/BUSI | 0.5556 | 0.3333 | 0.6667 | 0.1667 | 0.1111 | 0.3333 |
| ULTRASOUND/COVID-BLUES | 0.5000 | 0.0000 | 1.0000 | 0.0000 | 0.0000 | 0.0000 |
| ULTRASOUND/COVID_US | 0.4400 | 0.3333 | 0.6667 | 0.0920 | 0.0533 | 0.3333 |
| MRI/BRAIN_TUMOR | 0.6354 | 0.2708 | 0.7569 | 0.1437 | 0.3502 | 0.2708 |
| MRI/BRAIN_TUMOR_2 | 0.6585 | 0.7083 | 0.7083 | 0.6483 | 0.7742 | 0.7083 |
| PATHOLOGY/BCSS | 0.6771 | 0.3542 | 0.7847 | 0.2464 | 0.2224 | 0.3542 |
| PATHOLOGY/LC25000 | 0.8467 | 0.6167 | 0.9042 | 0.5609 | 0.5281 | 0.6167 |
| **Modality Summaries** | | | | | | |
| CT | 0.5247 | 0.4067 | 0.5933 | 0.2360 | 0.1690 | 0.4067 |
| CHEST_XRAY | 0.6965 | 0.1790 | 0.8431 | 0.0965 | 0.0824 | 0.1790 |
| FUNDUS | 0.6690 | 0.2167 | 0.7833 | 0.0827 | 0.0512 | 0.2167 |
| DERM | 0.6403 | 0.3375 | 0.7187 | 0.2446 | 0.2419 | 0.3375 |
| MAMMO | 0.4664 | 0.3000 | 0.7000 | 0.0582 | 0.0332 | 0.3000 |
| ULTRASOUND | 0.4985 | 0.2222 | 0.7778 | 0.0862 | 0.0548 | 0.2222 |
| MRI | 0.6470 | 0.4896 | 0.7326 | 0.3960 | 0.5622 | 0.4896 |
| PATHOLOGY | 0.7619 | 0.4854 | 0.8444 | 0.4036 | 0.3752 | 0.4854 |
| **Overall Results** | | | | | | |
| Overall | 0.6130 | 0.3296 | 0.7492 | 0.2005 | 0.1962 | 0.3296 |

Table 11: **Performance metrics of 7B model trained with RLOO on Vision Data**

| Dataset | Accuracy | Sensitivity | Specificity | F1 | Precision | Recall |
|---|---|---|---|---|---|---|
| **Individual Datasets** | | | | | | |
| CT/INSPECT | 0.6835 | 0.2000 | 0.8000 | 0.0691 | 0.0417 | 0.2000 |
| CT/HEMORRHAGE | 0.5000 | 0.5000 | 0.5000 | 0.3333 | 0.2500 | 0.5000 |
| CT/KITS23 | 0.4103 | 0.5000 | 0.5417 | 0.2885 | 0.2027 | 0.5000 |
| CT/LNDB | 0.4583 | 0.4583 | 0.4583 | 0.4222 | 0.4444 | 0.4583 |
| CT/RSPECT | 0.5556 | 0.3333 | 0.6667 | 0.1667 | 0.1111 | 0.3333 |
| CHEST_XRAY/VINDR | 0.7514 | 0.0763 | 0.9305 | 0.0256 | 0.0248 | 0.0763 |
| CHEST_XRAY/MIMIC-CXR | 0.7626 | 0.0942 | 0.9396 | 0.0495 | 0.0586 | 0.0942 |
| CHEST_XRAY/CHEXPERT_FULL | 0.7583 | 0.1099 | 0.9370 | 0.1032 | 0.1286 | 0.1099 |
| CHEST_XRAY/CORONAHACK | 0.5556 | 0.3333 | 0.6667 | 0.1667 | 0.1111 | 0.3333 |
| CHEST_XRAY/COVID19 | 0.6481 | 0.2500 | 0.7500 | 0.1143 | 0.0741 | 0.2500 |
| FUNDUS/JINCHI | 0.6250 | 0.2500 | 0.7500 | 0.1026 | 0.0645 | 0.2500 |
| FUNDUS/APTOS | 0.6933 | 0.2333 | 0.8083 | 0.1252 | 0.1702 | 0.2333 |
| FUNDUS/MESSIDOR-2 | 0.7021 | 0.2000 | 0.8000 | 0.0821 | 0.0516 | 0.2000 |
| DERM/HAM10000 | 0.6833 | 0.2083 | 0.8021 | 0.1077 | 0.1415 | 0.2083 |
| DERM/ISIC2020 | 0.5000 | 0.5000 | 0.5000 | 0.3333 | 0.2500 | 0.5000 |
| DERM/PAD_UFES_20 | 0.7208 | 0.2667 | 0.8195 | 0.2047 | 0.2509 | 0.2667 |
| MAMMO/CBIS | 0.6082 | 0.2000 | 0.7975 | 0.0083 | 0.0043 | 0.2000 |
| MAMMO/VINDR | 0.6800 | 0.2000 | 0.8000 | 0.0671 | 0.0403 | 0.2000 |
| MAMMO/CMMD | 0.1111 | 0.5000 | 0.5000 | 0.1000 | 0.0556 | 0.5000 |
| ULTRASOUND/BUSI | 0.6574 | 0.4861 | 0.7431 | 0.3829 | 0.3261 | 0.4861 |
| ULTRASOUND/COVID-BLUES | 0.5000 | 0.0000 | 1.0000 | 0.0000 | 0.0000 | 0.0000 |
| ULTRASOUND/COVID_US | 0.4400 | 0.3333 | 0.6667 | 0.0920 | 0.0533 | 0.3333 |
| MRI/BRAIN_TUMOR | 0.6406 | 0.2813 | 0.7604 | 0.1605 | 0.4819 | 0.2813 |
| MRI/BRAIN_TUMOR_2 | 0.6098 | 0.6667 | 0.6667 | 0.5900 | 0.7576 | 0.6667 |
| PATHOLOGY/BCSS | 0.6302 | 0.2604 | 0.7535 | 0.1208 | 0.3132 | 0.2604 |
| PATHOLOGY/LC25000 | 0.7900 | 0.4750 | 0.8688 | 0.3765 | 0.4051 | 0.4750 |
| **Modality Summaries** | | | | | | |
| CT | 0.5215 | 0.3983 | 0.5933 | 0.2559 | 0.2100 | 0.3983 |
| CHEST_XRAY | 0.6952 | 0.1728 | 0.8448 | 0.0918 | 0.0795 | 0.1728 |
| FUNDUS | 0.6735 | 0.2278 | 0.7861 | 0.1033 | 0.0954 | 0.2278 |
| DERM | 0.6347 | 0.3250 | 0.7072 | 0.2152 | 0.2141 | 0.3250 |
| MAMMO | 0.4664 | 0.3000 | 0.6992 | 0.0585 | 0.0334 | 0.3000 |
| ULTRASOUND | 0.5325 | 0.2731 | 0.8032 | 0.1583 | 0.1265 | 0.2731 |
| MRI | 0.6252 | 0.4740 | 0.7135 | 0.3752 | 0.6197 | 0.4740 |
| PATHOLOGY | 0.7101 | 0.3677 | 0.8111 | 0.2487 | 0.3591 | 0.3677 |
| **Overall Results** | | | | | | |
| Overall | 0.6074 | 0.3173 | 0.7448 | 0.1884 | 0.2172 | 0.3173 |

Table 12: **Performance metrics of 7B model trained with Reinforce++ on Vision Data**

| Dataset | Accuracy | Sensitivity | Specificity | F1 | Precision | Recall |
|---|---|---|---|---|---|---|
| **Individual Datasets** | | | | | | |
| CT/INSPECT | 0.6835 | 0.2000 | 0.8000 | 0.0691 | 0.0417 | 0.2000 |
| CT/HEMORRHAGE | 0.5000 | 0.5000 | 0.5000 | 0.3333 | 0.2500 | 0.5000 |
| CT/KITS23 | 0.3846 | 0.3667 | 0.5833 | 0.2391 | 0.1774 | 0.3667 |
| CT/LNDB | 0.5000 | 0.5000 | 0.5000 | 0.3333 | 0.2500 | 0.5000 |
| CT/RSPECT | 0.5556 | 0.3333 | 0.6667 | 0.1667 | 0.1111 | 0.3333 |
| CHEST_XRAY/VINDR | 0.7362 | 0.0811 | 0.9201 | 0.0087 | 0.0449 | 0.0811 |
| CHEST_XRAY/MIMIC-CXR | 0.7471 | 0.0930 | 0.9355 | 0.0414 | 0.1817 | 0.0930 |
| CHEST_XRAY/CHEXPERT_FULL | 0.7482 | 0.1269 | 0.9362 | 0.0922 | 0.2212 | 0.1269 |
| CHEST_XRAY/CORONAHACK | 0.5556 | 0.3333 | 0.6667 | 0.1667 | 0.1111 | 0.3333 |
| CHEST_XRAY/COVID19 | 0.6481 | 0.2500 | 0.7500 | 0.1143 | 0.0741 | 0.2500 |
| FUNDUS/JINCHI | 0.6250 | 0.2500 | 0.7500 | 0.1000 | 0.0625 | 0.2500 |
| FUNDUS/APTOS | 0.6833 | 0.2083 | 0.8021 | 0.0829 | 0.0914 | 0.2083 |
| FUNDUS/MESSIDOR-2 | 0.7021 | 0.2000 | 0.8000 | 0.0814 | 0.0511 | 0.2000 |
| DERM/HAM10000 | 0.6933 | 0.2333 | 0.8083 | 0.1515 | 0.4133 | 0.2333 |
| DERM/ISIC2020 | 0.5313 | 0.5208 | 0.5417 | 0.3829 | 0.7609 | 0.5208 |
| DERM/PAD_UFES_20 | 0.6981 | 0.2167 | 0.8052 | 0.1845 | 0.2750 | 0.2167 |
| MAMMO/CBIS | 0.6082 | 0.2000 | 0.7957 | 0.0086 | 0.0044 | 0.2000 |
| MAMMO/VINDR | 0.6800 | 0.2000 | 0.8000 | 0.0671 | 0.0403 | 0.2000 |
| MAMMO/CMMD | 0.1111 | 0.5000 | 0.5000 | 0.1000 | 0.0556 | 0.5000 |
| ULTRASOUND/BUSI | 0.5648 | 0.3333 | 0.6806 | 0.1702 | 0.1143 | 0.3333 |
| ULTRASOUND/COVID-BLUES | 0.5000 | 0.0000 | 1.0000 | 0.0000 | 0.0000 | 0.0000 |
| ULTRASOUND/COVID_US | 0.4400 | 0.3333 | 0.6667 | 0.0920 | 0.0533 | 0.3333 |
| MRI/BRAIN_TUMOR | 0.6250 | 0.2500 | 0.7500 | 0.1008 | 0.0632 | 0.2500 |
| MRI/BRAIN_TUMOR_2 | 0.4146 | 0.5000 | 0.5000 | 0.2931 | 0.2073 | 0.5000 |
| PATHOLOGY/BCSS | 0.6667 | 0.3333 | 0.7778 | 0.2424 | 0.4238 | 0.3333 |
| PATHOLOGY/LC25000 | 0.7367 | 0.3417 | 0.8354 | 0.2108 | 0.2386 | 0.3417 |
| **Modality Summaries** | | | | | | |
| CT | 0.5247 | 0.3800 | 0.6100 | 0.2283 | 0.1661 | 0.3800 |
| CHEST_XRAY | 0.6871 | 0.1769 | 0.8417 | 0.0846 | 0.1266 | 0.1769 |
| FUNDUS | 0.6702 | 0.2194 | 0.7840 | 0.0881 | 0.0683 | 0.2194 |
| DERM | 0.6409 | 0.3236 | 0.7184 | 0.2396 | 0.4831 | 0.3236 |
| MAMMO | 0.4664 | 0.3000 | 0.6986 | 0.0586 | 0.0334 | 0.3000 |
| ULTRASOUND | 0.5016 | 0.2222 | 0.7824 | 0.0874 | 0.0559 | 0.2222 |
| MRI | 0.5198 | 0.3750 | 0.6250 | 0.1970 | 0.1352 | 0.3750 |
| PATHOLOGY | 0.7017 | 0.3375 | 0.8066 | 0.2266 | 0.3312 | 0.3375 |
| **Overall Results** | | | | | | |
| Overall | 0.5890 | 0.2918 | 0.7333 | 0.1513 | 0.1750 | 0.2918 |

Table 13: **Performance metrics of 7B model trained with ReMax on Vision Data**

| Dataset | Accuracy | Sensitivity | Specificity | F1 | Precision | Recall |
|---|---|---|---|---|---|---|
| **Individual Datasets** | | | | | | |
| CT/INSPECT | 0.6835 | 0.2000 | 0.8000 | 0.0691 | 0.0417 | 0.2000 |
| CT/HEMORRHAGE | 0.5000 | 0.5000 | 0.5000 | 0.3333 | 0.2500 | 0.5000 |
| CT/KITS23 | 0.3846 | 0.4667 | 0.5208 | 0.2692 | 0.1892 | 0.4667 |
| CT/LNDB | 0.5000 | 0.5000 | 0.5000 | 0.3333 | 0.2500 | 0.5000 |
| CT/RSPECT | 0.5556 | 0.3333 | 0.6667 | 0.1667 | 0.1111 | 0.3333 |
| CHEST_XRAY/VINDR | 0.7396 | 0.0694 | 0.9244 | 0.0057 | 0.0030 | 0.0694 |
| CHEST_XRAY/MIMIC-CXR | 0.7548 | 0.0923 | 0.9376 | 0.0450 | 0.0442 | 0.0923 |
| CHEST_XRAY/CHEXPERT_FULL | 0.7456 | 0.1218 | 0.9376 | 0.0832 | 0.0665 | 0.1218 |
| CHEST_XRAY/CORONAHACK | 0.5556 | 0.3333 | 0.6667 | 0.1667 | 0.1111 | 0.3333 |
| CHEST_XRAY/COVID19 | 0.6481 | 0.2500 | 0.7500 | 0.1143 | 0.0741 | 0.2500 |
| FUNDUS/JINCHI | 0.6250 | 0.2500 | 0.7500 | 0.1008 | 0.0632 | 0.2500 |
| FUNDUS/APTOS | 0.6800 | 0.2000 | 0.8000 | 0.0671 | 0.0403 | 0.2000 |
| FUNDUS/MESSIDOR-2 | 0.7021 | 0.2000 | 0.8000 | 0.0814 | 0.0511 | 0.2000 |
| DERM/HAM10000 | 0.6833 | 0.2083 | 0.8021 | 0.1079 | 0.1340 | 0.2083 |
| DERM/ISIC2020 | 0.5625 | 0.5625 | 0.5625 | 0.4589 | 0.7667 | 0.5625 |
| DERM/PAD_UFES_20 | 0.7094 | 0.2417 | 0.8129 | 0.1550 | 0.3262 | 0.2417 |
| MAMMO/CBIS | 0.6122 | 0.2083 | 0.8008 | 0.0242 | 0.2042 | 0.2083 |
| MAMMO/VINDR | 0.6800 | 0.2000 | 0.8000 | 0.0671 | 0.0403 | 0.2000 |
| MAMMO/CMMD | 0.1111 | 0.5000 | 0.5000 | 0.1000 | 0.0556 | 0.5000 |
| ULTRASOUND/BUSI | 0.6481 | 0.4722 | 0.7361 | 0.3810 | 0.3300 | 0.4722 |
| ULTRASOUND/COVID-BLUES | 0.5000 | 0.0000 | 1.0000 | 0.0000 | 0.0000 | 0.0000 |
| ULTRASOUND/COVID_US | 0.4667 | 0.3636 | 0.6825 | 0.1508 | 0.3889 | 0.3636 |
| MRI/BRAIN_TUMOR | 0.6510 | 0.3021 | 0.7674 | 0.1929 | 0.4841 | 0.3021 |
| MRI/BRAIN_TUMOR_2 | 0.4146 | 0.5000 | 0.5000 | 0.2931 | 0.2073 | 0.5000 |
| PATHOLOGY/BCSS | 0.6771 | 0.3542 | 0.7847 | 0.2575 | 0.4475 | 0.3542 |
| PATHOLOGY/LC25000 | 0.7533 | 0.3833 | 0.8458 | 0.2903 | 0.3582 | 0.3833 |
| **Modality Summaries** | | | | | | |
| CT | 0.5247 | 0.4000 | 0.5975 | 0.2343 | 0.1684 | 0.4000 |
| CHEST_XRAY | 0.6887 | 0.1734 | 0.8433 | 0.0830 | 0.0598 | 0.1734 |
| FUNDUS | 0.6690 | 0.2167 | 0.7833 | 0.0831 | 0.0515 | 0.2167 |
| DERM | 0.6518 | 0.3375 | 0.7258 | 0.2406 | 0.4089 | 0.3375 |
| MAMMO | 0.4678 | 0.3028 | 0.7003 | 0.0638 | 0.1000 | 0.3028 |
| ULTRASOUND | 0.5383 | 0.2786 | 0.8062 | 0.1772 | 0.2396 | 0.2786 |
| MRI | 0.5328 | 0.4010 | 0.6337 | 0.2430 | 0.3457 | 0.4010 |
| PATHOLOGY | 0.7152 | 0.3688 | 0.8153 | 0.2739 | 0.4028 | 0.3688 |
| **Overall Results** | | | | | | |
| Overall | 0.5985 | 0.3098 | 0.7382 | 0.1749 | 0.2221 | 0.3098 |

Table 14: **Performance metrics of 7B model trained with PPO on Vision Data**

| Dataset | Accuracy | Sensitivity | Specificity | F1 | Precision | Recall |
|---|---|---|---|---|---|---|
| **Individual Datasets** | | | | | | |
| CT/INSPECT | 0.6835 | 0.2000 | 0.8000 | 0.0691 | 0.0417 | 0.2000 |
| CT/HEMORRHAGE | 0.5000 | 0.5000 | 0.5000 | 0.3333 | 0.2500 | 0.5000 |
| CT/KITS23 | 0.6154 | 0.5000 | 0.5000 | 0.3810 | 0.3077 | 0.5000 |
| CT/LNDB | 0.5000 | 0.5000 | 0.5000 | 0.3333 | 0.2500 | 0.5000 |
| CT/RSPECT | 0.5556 | 0.3333 | 0.6667 | 0.1667 | 0.1111 | 0.3333 |
| CHEST_XRAY/VINDR | 0.7327 | 0.0833 | 0.9166 | 0.0063 | 0.0033 | 0.0833 |
| CHEST_XRAY/MIMIC-CXR | 0.7242 | 0.0751 | 0.9298 | 0.0117 | 0.0752 | 0.0751 |
| CHEST_XRAY/CHEXPERT_FULL | 0.6908 | 0.0784 | 0.9233 | 0.0193 | 0.0476 | 0.0784 |
| CHEST_XRAY/CORONAHACK | 0.5556 | 0.3333 | 0.6667 | 0.1667 | 0.1111 | 0.3333 |
| CHEST_XRAY/COVID19 | 0.6481 | 0.2500 | 0.7500 | 0.1143 | 0.0741 | 0.2500 |
| FUNDUS/JINCHI | 0.6250 | 0.2500 | 0.7500 | 0.1000 | 0.0625 | 0.2500 |
| FUNDUS/APTOS | 0.6800 | 0.2000 | 0.8000 | 0.0671 | 0.0403 | 0.2000 |
| FUNDUS/MESSIDOR-2 | 0.7021 | 0.2000 | 0.8000 | 0.0814 | 0.0511 | 0.2000 |
| DERM/HAM10000 | 0.7217 | 0.3000 | 0.8271 | 0.2075 | 0.3442 | 0.3000 |
| DERM/ISIC2020 | 0.5313 | 0.5000 | 0.5625 | 0.3782 | 0.7614 | 0.5000 |
| DERM/PAD_UFES_20 | 0.7509 | 0.3333 | 0.8390 | 0.2482 | 0.2643 | 0.3333 |
| MAMMO/CBIS | 0.6449 | 0.2750 | 0.8051 | 0.0770 | 0.0640 | 0.2750 |
| MAMMO/VINDR | 0.6800 | 0.2000 | 0.8000 | 0.0667 | 0.0400 | 0.2000 |
| MAMMO/CMMD | 0.8889 | 0.5000 | 0.5000 | 0.4706 | 0.4444 | 0.5000 |
| ULTRASOUND/BUSI | 0.5556 | 0.3333 | 0.6667 | 0.1667 | 0.1111 | 0.3333 |
| ULTRASOUND/COVID-BLUES | 0.5000 | 0.0000 | 1.0000 | 0.0000 | 0.0000 | 0.0000 |
| ULTRASOUND/COVID_US | 0.4133 | 0.2500 | 0.6381 | 0.0741 | 0.0435 | 0.2500 |
| MRI/BRAIN_TUMOR | 0.6563 | 0.3125 | 0.7708 | 0.2030 | 0.2817 | 0.3125 |
| MRI/BRAIN_TUMOR_2 | 0.8780 | 0.8958 | 0.8958 | 0.8778 | 0.8864 | 0.8958 |
| PATHOLOGY/BCSS | 0.6276 | 0.2500 | 0.7535 | 0.1008 | 0.0632 | 0.2500 |
| PATHOLOGY/LC25000 | 0.8633 | 0.6583 | 0.9146 | 0.6280 | 0.7043 | 0.6583 |
| **Modality Summaries** | | | | | | |
| CT | 0.5709 | 0.4067 | 0.5933 | 0.2567 | 0.1921 | 0.4067 |
| CHEST_XRAY | 0.6703 | 0.1640 | 0.8373 | 0.0637 | 0.0622 | 0.1640 |
| FUNDUS | 0.6690 | 0.2167 | 0.7833 | 0.0828 | 0.0513 | 0.2167 |
| DERM | 0.6680 | 0.3778 | 0.7429 | 0.2780 | 0.4566 | 0.3778 |
| MAMMO | 0.7379 | 0.3250 | 0.7017 | 0.2048 | 0.1828 | 0.3250 |
| ULTRASOUND | 0.4896 | 0.1944 | 0.7683 | 0.0802 | 0.0515 | 0.1944 |
| MRI | 0.7671 | 0.6042 | 0.8333 | 0.5404 | 0.5840 | 0.6042 |
| PATHOLOGY | 0.7455 | 0.4542 | 0.8340 | 0.3644 | 0.3837 | 0.4542 |
| **Overall Results** | | | | | | |
| Overall | 0.6648 | 0.3429 | 0.7618 | 0.2339 | 0.2456 | 0.3429 |

