# OpenReview forum: "QoQ-Med: Building Multimodal Clinical Foundation Models with Domain-Aware GRPO Training"
_NeurIPS.cc/2025/Conference — NeurIPS 2025 oral_

### Official Review · Reviewer_mAXD · 2025-07-02

**Clarity:** 3
**Significance:** 3
**Originality:** 3
**Rating:** 6
**Confidence:** 4

**Summary:**

This paper presents QoQ-Med-7B/32B, an open-source multimodal clinical foundation model that jointly processes medical images (2D/3D), time-series signals (ECG), and text reports. The key contribution is Domain-aware Relative Policy Optimization (DRPO), a hierarchical reinforcement learning method that scales rewards based on domain rarity and task difficulty to address data imbalance in heterogeneous clinical datasets. The model is trained on 2.61M instruction-tuning pairs across 9 clinical domains and demonstrates significant improvements over existing methods.

**Questions:**

1. Can you provide confidence intervals through bootstrap sampling of your existing results? What is the variance across different random seeds for at least a subset of experiments?
2. What is the isolated contribution of cluster-level scaling? How sensitive are results to the number of clusters and reward weights?
3. How does DRPO perform on the rarest conditions in your datasets? Does the hierarchical weighting risk overfitting to noise in extremely small domains?

**Ethical Concerns:**

["NO or VERY MINOR ethics concerns only"]

**Final Justification:**

Thank you for your rebuttal. I have ajusted my rating accordingly.

**Limitations:**

The authors briefly acknowledge the limited sample efficiency of unsupervised reasoning learning in their conclusion, but a more comprehensive discussion of limitations is warranted. The most critical limitation is the lack of statistical validation - all experiments are single-run due to computational constraints, preventing any assessment of variance or statistical significance of the reported improvements. This is particularly concerning given that some performance gains are modest and could potentially fall within noise margins. The paper also lacks analysis of potential biases that DRPO's hierarchical weighting might introduce or amplify. By up-weighting underrepresented domains, the method could inadvertently overfit to spurious correlations in small datasets or amplify existing demographic biases present in those domains. The computational requirements (8xA100 GPUs for multiple days of training) severely limit accessibility and reproducibility, yet the environmental impact is not discussed. Additionally, while the model generates reasoning traces, there's no systematic evaluation of their clinical accuracy beyond selected examples, and no prospective clinical validation is provided. The reliance on pseudo-labels from BiomedParse for segmentation evaluation introduces another unvalidated source of error. Finally, the heavy dependence on appendices for critical implementation details means that key aspects of the method cannot be fully evaluated within the peer-review process, potentially hiding important limitations or implementation choices that could affect reproducibility and generalization.

**Paper Formatting Concerns:**

Not observed.

**Quality:**

3

**Strengths And Weaknesses:**

Strengths:
1. DRPO is a well-motivated extension of GRPO with clear theoretical foundation. The hierarchical scaling mechanism (domain-level and cluster-level) effectively addresses the critical problem of performance imbalance in multi-domain clinical learning. The mathematical formulation is rigorous and the algorithm design is sound.
2. First open-source model to successfully integrate 1D time-series (ECG) with 2D/3D medical images, addressing a significant gap in existing medical MLLMs. The architecture elegantly handles missing modalities.
3. The model generates reasoning traces and bounding boxes for localization, crucial for clinical trust. Qualitative examples with clinician annotations demonstrate practical utility, though more systematic evaluation is needed.

Weaknesses:
1. The paper's most significant limitation is the lack of statistical significance testing. All experiments are single-run due to computational constraints (acknowledged in checklist Q7), meaning no error bars or confidence intervals are provided. This fundamentally limits our confidence in the reported improvements, especially given that some gains are relatively modest. The paper also lacks systematic error analysis beyond a few qualitative examples. While Figure 4 shows some failure cases, there's no aggregate analysis of error types, failure modes, or when DRPO provides the most benefit versus baseline methods. The evaluation would be significantly strengthened by at least bootstrap-based confidence intervals on existing results and a more thorough characterization of model failures.
2. Despite working with clinical data where fairness is critical, the paper provides no analysis of demographic or institutional bias. It's unclear whether DRPO's domain weighting helps reduce disparities or potentially amplifies existing biases by over-emphasizing certain underrepresented domains that may correlate with demographic factors. The paper also doesn't evaluate performance on truly rare conditions beyond the main imaging modalities, leaving questions about robustness in extreme low-resource settings. Without subgroup performance analysis or evaluation on out-of-distribution data, the clinical safety and equity implications remain unclear.
3. The reasoning quality evaluation, while featuring clinician annotations, lacks methodological rigor. The paper doesn't specify how many clinicians participated, what their qualifications were, or what the inter-annotator agreement was. The evaluation appears to be purely qualitative on selected examples rather than systematic across the validation set. Furthermore, the use of BiomedParse model to generate pseudo-segmentation labels when ground truth is unavailable introduces another source of potential error that isn't validated. The heavy reliance on appendices for critical implementation details also makes it difficult to fully assess the methodology, as appendix content isn't peer-reviewed.

---

> ### Author Rebuttal · Authors · 2025-07-31
>
> We thank Reviewer mAXD for finding our method interesting, the mathematical formulation rigorous and the multimodal architecture novel.
>
> W1/Q1: We further ran the main experiments 4 times under each setting for statistical analysis, and included the updated results below:
>
> | Model | CXR |  | Mammo. |  | Dermoscopy |  | CT Scan |  |
> |---|---:|---:|---:|---:|---:|---:|---:|---:|
> | | **Acc** | **F1** | **Acc** | **F1** | **Acc** | **F1** | **Acc** | **F1** |
> | **SFT** | .688±.030 | .078±.013 | .481±.015 | .056±.004 | .640±.029 | .158±.000 | .525±.000 | .236±.000 |
> | **ReMax** | .636±.049 | .120±.030 | .577±.063 | .033±.020 | .644±.009 | .257±.018 | .567±.026 | .228±.001 |
> | **RE++** | .730±.027 | .082±.032 | .660±.087 | .076±.011 | .635±.008 | .237±.079 | .529±.013 | .247±.039 |
> | **RLOO** | **.752±.015** | .086±.003 | .471±.002 | .068±.006 | .636±.008 | .216±.028 | .534±.014 | .224±.042 |
> | **GRPO** | .703±.012 | .095±.007 | .466±.000 | .059±.000 | .646±.003 | .244±.016 | .524±.002 | .236±.001 |
> | **DRPO**_DomainOnly_ | .693±.001 | .086±.001 | .751±.008 | .213±.005 | .679±.016 | .251±.042 | .571±.000 | .257±.000 |
> | **DRPO** | .687±.005 | **.115±.014** | **.756±.008** | **.253±.006** | **.715±.006** | **.407±.006** | **.570±.012** | **.309±.013** |
>
> | Model | Fundus |  | Ultrasound |  | MRI |  | Pathology |  | Overall |  |
> |---|---:|---:|---:|---:|---:|---:|---:|---:|---:|---:|
> | | **Acc** | **F1** | **Acc** | **F1** | **Acc** | **F1** | **Acc** | **F1** | **Acc** | **F1** |
> | **SFT** | **.715±.051** | .066±.010 | .548±.028 | **.235±.000** | .567±.038 | .197±.000 | .652±.000 | .083±.000 | .602±.014 | .139±.001 |
> | **ReMax** | .678±.015 | .089±.006 | .547±.044 | .147±.008 | .547±.012 | .264±.018 | .706±.007 | .270±.011 | .596±.002 | .176±.001 |
> | **RE++** | .672±.003 | .098±.013 | .519±.010 | .136±.012 | .651±.022 | .420±.025 | .668±.024 | .254±.026 | .621±.020 | .202±.053 |
> | **RLOO** | .670±.003 | **.099±.013** | .519±.012 | .144±.014 | .658±.032 | .432±.020 | .699±.011 | .216±.024 | .611±.002 | .189±.003 |
> | **GRPO** | .670±.001 | .086±.002 | .520±.009 | .146±.011 | .631±.027 | .395±.030 | .715±.004 | .286±.010 | .609±.005 | .193±.003 |
> | **DRPO**_DomainOnly_ | .669±.000 | .083±.000 | .480±.040 | .098±.031 | .733±.012 | .475±.012 | **.762±.009** | **.388±.028** | .668±.004 | .237±.004 |
> | **DRPO** | .672±.004 | .093±.008 | **.555±.014** | .223±.021 | **.789±.019** | **.625±.028** | .708±.010 | .265±.016 | **.666±.009** | **.295±.013** |
>
>
> The standard deviation for all methods is around ~1% across runs. In terms of overall performance, our method (DRPO) significantly outperformed all baseline methods (SFT, ReMax, RE++, RLOO and GRPO) with p<0.0001.
>
> W2: We evaluated the fairness of the following eight datasets from CLIMB, where the demographic information is available: CheXpert, Vindr-Mammography, HAM10000, Hemorrhage, COVID-BLUES, ISIC2020, PAD-UFES-20. In particular, these datasets contain the age, gender and parent birthplace of the participants. We bin the age into 20-year brackets, treat each demographic information as a group, and evaluate the standard deviation of the accuracy and F1 scores across different groups. The results are as follows (lower is better):
>
> | Method | Age Std_acc | Age Std_f1 | Gender Std_acc | Gender Std_f1 | Parent Std_acc | Parent Std_f1 | Overall Std_acc | Overall Std_f1 |
> |--------|-------------|------------|----------------|---------------|----------------|---------------|-----------------|----------------|
> | GRPO   | **0.0290**  | 0.0345     | **0.0124**     | 0.0207        | 0.2955         | **0.2958**    | **0.1123**      | 0.1170         |
> | DRPO   | 0.0298      | **0.0279** | 0.0164         | **0.0154**    | **0.2927**     | 0.3023        | 0.1130          | **0.1152**     |
>
> In general, we do not observe a significant difference in fairness between GRPO and DRPO (our method). Our method gives a more homogeneous F1 score across groups, while the standard deviation in accuracy across groups for GRPO is lower. We will conduct a more thorough analysis of the fairness of these models and training methods as a part of our ongoing effort.
>
> Q2: The number of clusters in the model is determined automatically via the elbow method, as described in Sec. 3.3 and App. B. 2, with the possibility to set an upper limit on the number of clusters. In the original experiment, the cluster limit for each domain is set to 10. We tested the model with 1 (no clustering), 3, 10 and 20 clusters. The results are as follows:
>
>
> | **Cluster** | **CXR** |  | **Mam** |  | **Derm** |  | **CT** |  | **Fund** |  | **US** |  | **MRI** |  | **Path** |  | **All** |  |
> |-----|-----|-----|-----|-----|-----|-----|-----|-----|-----|-----|-----|-----|-----|-----|-----|-----|-----|-----|
> |     | Acc | F1 | Acc | F1 | Acc | F1 | Acc | F1 | Acc | F1 | Acc | F1 | Acc | F1 | Acc | F1 | Acc | F1 |
> | **1** | **.694**| .085 | .746 | .211 | .678 | .286 | .571 | .257 | .669 | .083 | .544 | .200 | .757 | .505 | **.773**| **.449**| .679 | .259 |
> | **3** | **.694**| .125 | .568 | .048 | .680 | .356 | .562 | .284 | **.672**| .147 | .520 | .152 | .717 | .546 | .723 | .289 | .642 | .244 |
> | **10** | .691 | .125 | **.759**| .253 | **.707**| **.400**| **.580**| **.321**| .670 | .088 | **.568**| **.240**| **.806**| **.652**| .707 | .303 | **.686**| .286 |
> | **20** | .668 | **.167**| .751 | **.268**| .675 | .300 | .548 | .262 | .635 | **.166**| .547 | .214 | .804 | .649 | .731 | .329 | .670 | **.294**|
>
> In general, we observe that having no cluster or a very low cluster limit will cause a decrease in performance. A higher cluster limit, however, does not seem to hurt the performance, as the elbow method automatically chooses a lower cluster count than the limit. This allows the algorithm to remain efficient under arbitrary cluster limits.
>
> Q3: The core contribution of the hierarchical scaling scheme is that it performs better for rare conditions and rare datasets. As described in Figure 2, compared to GRPO, DRPO performs particularly better in clinical modalities where data is rare. In addition, the gap between training and validation accuracy is similar across all methods and modalities, which indicates that DRPO does not cause the model to overfit to smaller datasets.
>
> W3: We agree that synthetic segmentation labels may introduce errors for some rare cases.  BiomedParse reported an average dice score of 0.91-0.98 across clinical domains. We randomly selected 100 images evenly across the eight imaging domains where ground truth segmentations are not available. We observed that the segmentations for most classes are highly consistent with the true abnormalities, like Lung Opacity, Acute pulmonary embolism and Pituitary Tumor, with an estimated dice score of >0.8. However, the segmentations for some rarely seen or novel classes, like COVID-19 and malignant lesions in breast ultrasound, are less accurate, with an estimated dice score in the 0.3-0.4 range.  Due to the restriction of visual content in the rebuttal, we will provide a detailed visual and quantitative analysis of the generated segmentations in the final manuscript.
>
> In the final manuscript, we will discuss the environmental impact in App. D.1. Our A100 SXM has a TDP of 400W. Assuming CPUs and other components have a combined TDP of 600W, training of a 7B model requires an electricity of 182.4kWh. Under the same assumption, training a 32B model requires an electricity of 571.2 kWh. We would also like to point out that releasing public models as we did has a positive environmental impact, as researchers can directly use or finetune the model on the domain data, instead of doing the pretraining repeatedly.

---

> ### Author Response · Authors · 2025-08-07
> **Thanks for the review**
>
> Thanks a lot for reviewing our paper and giving constructive feedbacks! We hope our rebuttal has addressed your concerns. As the discussion period is about to end, please let us know if you have any other questions that remains to be addressed.

---

### Official Review · Reviewer_L5Uq · 2025-07-03

**Clarity:** 4
**Significance:** 4
**Originality:** 4
**Rating:** 5
**Confidence:** 5

**Summary:**

This paper introduces QoQ-Med, a novel multimodal foundation model for clinical applications, uniquely capable of reasoning over diverse data types including medical images (2D/3D), time-series signals (ECG), and text. The core technical contribution is a new reinforcement learning algorithm, Domain-aware Group Relative Policy Optimization (DRPO). DRPO is designed to address the critical challenge of data heterogeneity and imbalance in clinical datasets by introducing a hierarchical reward scaling mechanism. The authors demonstrate that DRPO significantly improves diagnostic performance over standard RL methods, particularly in underrepresented domains. The resulting model, which generates both a final diagnosis and interpretable outputs like reasoning traces and salient region highlighting, is shown to be competitive with or superior to existing open-source and even some closed-source models.

**Questions:**

N/A

**Ethical Concerns:**

["NO or VERY MINOR ethics concerns only"]

**Final Justification:**

Thank you for the rich dialogue, In overall, I am in favour for the acceptance of the paper.

**Limitations:**

A minor limitation is that while the model handles multiple modalities, the fusion mechanism is a relatively simple concatenation of projected tokens, a more sophisticated fusion mechanism might yield further improvements, though this is more of a future work direction than a flaw.

**Quality:**

4

**Strengths And Weaknesses:**

Strengths:
---

- The paper tackles a significant and challenging problem, which is creating a generalist clinical foundation model that handles truly heterogeneous data. The integration of 1D time-series data (ECG) with 2D/3D imaging and text is a notable step forward from vision-centric medical MLLMs and addresses a real clinical need for holistic patient data analysis.

- The proposed DRPO algorithm is a novel and well-motivated contribution. It provides a principled way to handle domain imbalance within a critic-free RL framework, which is computationally efficient. The hierarchical scaling based on both domain rarity and intra-domain difficulty is an intelligent design that directly addresses a known problem in training on real-world. The empirical results, showing up to a 43% macro-F1 score improvement, strongly support its effectiveness.

- The experimental evaluation is comprehensive and well-designed. The authors compare their method against a wide range of strong baselines (SFT, PPO ...). They evaluate on a large, multi-domain dataset and include specific experiments to validate multimodal fusion and interpretability. The ablation studies clearly demonstrate the contribution of each component of DRPO.

- Strong comparison against other publicly available systems in the Table 4.

- The authors plan to release the model weights (7B and 32B), the modular training pipeline and the extensive reasoning traces is a major strength. This significantly enhances reproducibility and will provide a valuable resource for the research community, lowering the barrier to entry for research.

Weaknesses:
---

N/A

---

> ### Author Rebuttal · Authors · 2025-07-31
>
> We thank reviewer L5Uq for finding our DRPO method interesting, our task important, and our experiments comprehensive. We are excited to see what researchers will build upon our released multimodal disease diagnosis models. We agree that a more sophisticated fusion mechanism may be helpful in aligning multiple modalities. There have been some recent efforts to tackle this challenge. [FUSION](https://arxiv.org/html/2504.09925v1), for example, utilizes similarity loss for the alignment between vision and text embeddings. And [EMMA](https://openreview.net/forum?id=lbrO3bGpeO) proposed a lightweight module for cross-modality alignment. Most of these efforts are centered around texts and images. In our cases, fusion of time series, vision and texts is a non-trivial challenge. While it is out of the scope of this work, we will explore this in our future work, and we welcome community efforts to build better multimodal representations.

---

### Official Review · Reviewer_pph3 · 2025-07-13

**Clarity:** 3
**Significance:** 3
**Originality:** 2
**Rating:** 5
**Confidence:** 3

**Summary:**

This paper presents QoQ-Med, a multimodal clinical foundation model that processes medical images (2D/3D), time-series signals (ECG), and text. The model is trained with Domain-aware Relative Policy Optimization (DRPO), a new reinforcement learning objective that extends GRPO. DRPO boosts diagnostic performance and hierarchically scales rewards based on domain rarity and example difficulty to balance training.

**Questions:**

1. Availability of the Reasoning Traces Dataset: The commitment to release the 2.61 million reasoning traces is a significant and commendable contribution that would greatly benefit the research community. However, upon reviewing the provided anonymous link, I was unable to locate this specific dataset. Could the authors please clarify the status of this resource or provide a direct pointer to it?
2. Analysis of Reasoning and Failure Modes: Figure 4 provides compelling qualitative examples of QoQ-Med's successful reasoning process, which is a key strength. To offer a more comprehensive and balanced assessment of the model's true capabilities and limitations, it would be highly beneficial to include an analysis of its common failure modes. For instance:
  - Are there recurring patterns when the generated reasoning is clinically incorrect or irrelevant?
  - Could the authors provide examples where the reasoning seems plausible but is factually disconnected from the input image?

**Ethical Concerns:**

["NO or VERY MINOR ethics concerns only"]

**Final Justification:**

My recommendation is to Accept.

I thank the authors for their detailed rebuttal. The responses have successfully addressed the main concerns and questions I raised in my initial review.

Resolved issues: the weight of each reward

Unresolved issues: none

**Limitations:**

yes

**Quality:**

3

**Strengths And Weaknesses:**

Strengths
1. Multimodality: The proposed model, QoQ-Med, is significant as it is one of the first open-source foundation models to jointly process 1D time-series data (ECG) alongside 2D/3D medical images and text.
2. Comprehensive evaluation: The experimental evaluation is thorough and convincing. The results show that DRPO achieves substantial performance improvements (e.g., a 43% increase in macro-F1) over established methods. The ablation studies effectively isolate and validate the contributions of DRPO's key components.
3. Writing: The paper is well-written and clearly structured. The methodology is explained logically, and Figure 1 provides an intuitive overview of both the model architecture, greatly enhancing the reader's understanding.

Weaknesses
1. Significance analysis: The primary weakness is a lack of statistical significance analysis for the experimental results. Given the high variance often seen in RL training, reporting error bars or confidence intervals from multiple runs would be crucial to substantiate the robustness of the claimed performance gains.
2. Hyperparameters: The design of the combined reward function involves several weighting hyperparameters (e.g., $\lambda_{acc}, \lambda_{IoU}, \lambda_{aux}$). The paper does not provide a sensitivity analysis or justification for the specific values chosen, leaving it unclear how performance might be affected by different choices.

---

> ### Author Rebuttal · Authors · 2025-07-31
>
> We thank Reviewer pph3 for finding our multimodality task important, evaluation comprehensive and writing well-structured.
>
>
> Weakness 1: We further ran the main experiments 4 times under each setting for statistical analysis, and included the updated results below:
>
>
> | Model | CXR |  | Mammo. |  | Dermoscopy |  | CT Scan |  |
> |---|---:|---:|---:|---:|---:|---:|---:|---:|
> | | **Acc** | **F1** | **Acc** | **F1** | **Acc** | **F1** | **Acc** | **F1** |
> | **SFT** | .688±.030 | .078±.013 | .481±.015 | .056±.004 | .640±.029 | .158±.000 | .525±.000 | .236±.000 |
> | **ReMax** | .636±.049 | .120±.030 | .577±.063 | .033±.020 | .644±.009 | .257±.018 | .567±.026 | .228±.001 |
> | **RE++** | .730±.027 | .082±.032 | .660±.087 | .076±.011 | .635±.008 | .237±.079 | .529±.013 | .247±.039 |
> | **RLOO** | **.752±.015** | .086±.003 | .471±.002 | .068±.006 | .636±.008 | .216±.028 | .534±.014 | .224±.042 |
> | **GRPO** | .703±.012 | .095±.007 | .466±.000 | .059±.000 | .646±.003 | .244±.016 | .524±.002 | .236±.001 |
> | **DRPO**_DomainOnly_ | .693±.001 | .086±.001 | .751±.008 | .213±.005 | .679±.016 | .251±.042 | .571±.000 | .257±.000 |
> | **DRPO** | .687±.005 | **.115±.014** | **.756±.008** | **.253±.006** | **.715±.006** | **.407±.006** | **.570±.012** | **.309±.013** |
>
> | Model | Fundus |  | Ultrasound |  | MRI |  | Pathology |  | Overall |  |
> |---|---:|---:|---:|---:|---:|---:|---:|---:|---:|---:|
> | | **Acc** | **F1** | **Acc** | **F1** | **Acc** | **F1** | **Acc** | **F1** | **Acc** | **F1** |
> | **SFT** | **.715±.051** | .066±.010 | .548±.028 | **.235±.000** | .567±.038 | .197±.000 | .652±.000 | .083±.000 | .602±.014 | .139±.001 |
> | **ReMax** | .678±.015 | .089±.006 | .547±.044 | .147±.008 | .547±.012 | .264±.018 | .706±.007 | .270±.011 | .596±.002 | .176±.001 |
> | **RE++** | .672±.003 | .098±.013 | .519±.010 | .136±.012 | .651±.022 | .420±.025 | .668±.024 | .254±.026 | .621±.020 | .202±.053 |
> | **RLOO** | .670±.003 | **.099±.013** | .519±.012 | .144±.014 | .658±.032 | .432±.020 | .699±.011 | .216±.024 | .611±.002 | .189±.003 |
> | **GRPO** | .670±.001 | .086±.002 | .520±.009 | .146±.011 | .631±.027 | .395±.030 | .715±.004 | .286±.010 | .609±.005 | .193±.003 |
> | **DRPO**_DomainOnly_ | .669±.000 | .083±.000 | .480±.040 | .098±.031 | .733±.012 | .475±.012 | **.762±.009** | **.388±.028** | .668±.004 | .237±.004 |
> | **DRPO** | .672±.004 | .093±.008 | **.555±.014** | .223±.021 | **.789±.019** | **.625±.028** | .708±.010 | .265±.016 | **.666±.009** | **.295±.013** |
>
>
> The standard deviation for all methods is around ~1% across runs. In terms of overall performance, our method (DRPO) significantly outperformed all baseline methods (SFT, ReMax, RE++, RLOO and GRPO) with p<0.0001.
>
> Weakness 2: In general, we found that the weight of each reward does not have a significant impact on the final performance. In particular, the auxiliary rewards on formatting saturate shortly in the early stages of training. They have effectively no impact on the later stages due to normalization. We tested different combinations of accuracy rewards: semantic alignment rewards. The results are as follows:
>
> | **Acc:IoU** | **CXR** |       | **Mammo.** |       | **Dermoscopy** |       | **CT Scan** |       |
> |-------------|---------|-------|------------|-------|----------------|-------|-------------|-------|
> |             | **Acc** | **F1** | **Acc**    | **F1** | **Acc**        | **F1** | **Acc**     | **F1** |
> | **0.6:0.2** | **.691**    | .125 | **.759**   | **.253**  | **.707**       | **.400** | **.580**        | **.321**  |
> | **0.2:0.6** | .690    | **.147**  | .563       | .185 | .668           | .290  | .576    | .308 |
>
> | **Acc:IoU** | **Fundus** |       | **Ultrasound** |       | **MRI** |       | **Pathology** |       | **Overall** |       |
> |-------------|------------|-------|----------------|-------|---------|-------|---------------|-------|-------------|-------|
> |             | **Acc**    | **F1** | **Acc**        | **F1** | **Acc** | **F1** | **Acc**       | **F1** | **Acc**     | **F1** |
> | **0.6:0.2** | .670       | .088  | .568       | **.240** | **.806** | **.652** | **.707**          | **.303**  | **.686**    | **.286** |
> | **0.2:0.6** | **.681**   | **.136** | **.573**           | .218  | .768    | .561  | .698      | .233 | .652        | .260  |
>
> Decreasing the weight of the accuracy reward gives a drop in overall performance and performance in most domains, but results are still significantly better than all baselines, which demonstrates the robustness of DRPO.
>
> Q1: We included a sample reasoning trace in the anonymous repo under datasets/. Due to the size limit of the anonymous repo and the restriction on external links, we are not able to share the entire reasoning traces now, but we will include them in the final manuscript.
>
> Q2/3: We included examples of reasoning traces that lead to incorrect answers in App. E and Sec. 4.3. Generally, there are 2 areas where the current model seems to struggle. First, the model seems to struggle to spot minor objects like support devices. In addition, the model was not particularly good at diagnosing novel diseases like COVID from ultrasound, where the most common misdiagnosis is pneumonia. From our randomly selected samples, there is no instance where the reasoning evidence is not aligned with the image. We will conduct a more thorough error analysis in the appendix of the final manuscript.

---

> > ### Comment · Reviewer_pph3 · 2025-08-01
> > **Response to Rebuttal**
> >
> > I thank the authors for their detailed rebuttal. The responses have successfully addressed the main concerns and questions I raised in my initial review.

---

> > > ### Author Response · Authors · 2025-08-04
> > > **Thanks for the feedback**
> > >
> > > Thank you for recognizing that we have addressed your concerns. We greatly appreciate your constructive feedback throughout the review process.

---

### Official Review · Reviewer_MhFk · 2025-07-19

**Clarity:** 3
**Significance:** 3
**Originality:** 4
**Rating:** 5
**Confidence:** 3

**Summary:**

This paper introduces a QoQ-Med model, a large Multimodal Clinical Foundation Model with Domain-Aware GRPO Training. The model takes medical images, time-series signals, and text reports as inputs. It also introduces a novel reinforcement learning method, Domain-aware Relative Policy Optimization (DRPO), which adjusts rewards based on domain rarity and modality difficulty to address data imbalance. The model is trained on 2.61 million instruction-tuning samples from 9 clinical domains. Experimental results show strong performance.

**Questions:**

1. The logic in clinical diagnosis varies significantly in different fields. Please discuss more on the applicability and adaptability of QoQ-style training across domains with differing reasoning structures.
2. Although the model supports images, text, and time-series, there is little analysis on how easily the framework scales to additional modalities (e.g., pathology slides, genomic data, 3D scans). Please discuss how to extend the model on further modality data/inputs.

**Ethical Concerns:**

["NO or VERY MINOR ethics concerns only"]

**Limitations:**

Yes

**Quality:**

3

**Strengths And Weaknesses:**

Strengths:
1. The multimodal inputs setup closely mirrors real clinical diagnostic reasoning, going beyond standard VQA
2. The model is trained on high-quality data, including 2.61 million question-answer pairs with intermediate reasoning
3. The method employs the SOTA GRPO in the model training
4. Experimental results look very promising.

Weaknesses:
1. The logic in clinical diagnosis varies significantly in different fields. Please discuss more on the applicability and adaptability of QoQ-style training across domains with differing reasoning structures.
2. The study lacks expert human evaluation or annotation, such as rating answer helpfulness or diagnostic correctness.
3. Although the model supports images, text, and time-series, there is little analysis on how easily the framework scales to additional modalities (e.g., pathology slides, genomic data, 3D scans).

---

> ### Author Rebuttal · Authors · 2025-07-31
>
> We thank Reviewer MhFk for the constructive feedback and suggestions.
>
>
> W1/Q1: The reasoning training from QoQ style training is highly generalizable, as we do not presume any reasoning structure during the training process. The supervision is only on the final diagnosis. This follows the finding from the [Deepseek R1](https://arxiv.org/abs/2501.12948) paper, where the authors demonstrated how simple, flexible supervision elicits models’ pretraining knowledge, helps prevent reward hacking and enables more holistic learning. In Sec. 4.3 and App. E, we detailed how the model reasons across different modalities. In particular, we found that the model is able to focus more on the color for the reasoning of dermoscopy, with examples like “slightly raised, pinkish, and slightly elevated area.” On the other hand, the model focuses more on the brightness, patterns and textures for ultrasound videos, with examples like “The hyperechoic nature of the lesion is often associated with consolidation.”
>
>
> W2: As described in Sec. 4.3, we collaborated with expert clinicians to annotate the relevance of the reasoning traces to the diagnosis. The expert annotation in App. E indicates that around 93.8% of the reasoning traces are relevant to the diagnosis of the disease. In particular, around 45.3% of the reasoning is rated as highly relevant, and 100% of the rated samples contain highly relevant reasoning traces.
>
> | Model | CXR (Acc/F1) | Mammo. (Acc/F1) | Dermoscopy (Acc/F1) | CT Scan (Acc/F1) | Fundus (Acc/F1) | Ultrasound (Acc/F1) | MRI (Acc/F1) | Pathology (Acc/F1) | Overall (Acc/F1) |
> |-------|--------------|-----------------|---------------------|------------------|-----------------|---------------------|--------------|--------------------|--------------------|
> | SFT | .679/.077 | .726/.182 | .624/.158 | .571/.257 | .669/.083 | .561/.235 | .520/.197 | .653/.083 | .625/.159 |
> | PPO | .670/.064 | .738/.205 | .668/.278 | .571/.257 | .669/.083 | .490/.080 | .767/.540 | .745/.364 | .665/.234 |
> | ReMax | .689/.083 | .468/.064 | .652/.241 | .525/.234 | .669/.083 | .538/.177 | .533/.243 | .715/.274 | .599/.175 |
> | RE++ | .687/.085 | .466/.059 | .641/.240 | .525/.228 | .670/.088 | .502/.087 | .520/.197 | .702/.227 | .589/.151 |
> | RLOO | .695/.092 | .466/.058 | .635/.215 | .522/.256 | .673/.103 | .532/.158 | .625/.375 | .710/.249 | .607/.188 |
> | GRPO | .696/.096 | .466/.058 | .640/.245 | .525/.236 | .669/.083 | .499/.086 | .647/.396 | .762/.404 | .613/.200 |
> | DRPO_DomainOnly | .694/.085 | .746/.211 | .678/.286 | .571/.257 | .669/.083 | .544/.200 | .757/.505 | .773/.449 | .679/.259 |
> | DRPO_NoKL | .685/.103 | .711/.264 | .691/.382 | .597/.365 | .676/.085 | .554/.228 | .722/.535 | .710/.300 | .668/.283 |
> | DRPO | .691/.125 | .759/.253 | .707/.400 | .580/.321 | .670/.088 | .568/.240 | .806/.652 | .707/.303 | .686/.286 |
>
> W3/Q2: We would like to clarify that our model QoQ-Med does support pathology slides, as well as 3D scans (CT, MRI) and videos (ultrasound). The performance is demonstrated in Table 2 (attached above for reference). For pathology slides, DRPO achieves an F1 score of 0.449, representing an 11.1% improvement over the next best baseline GRPO (0.404). In 3D imaging modalities, our method shows strong results on MRI with DRPO achieving 0.652 F1 score, a substantial 20.7% improvement over the next best baseline PPO's 0.540. For CT scans, DRPO without KL divergence demonstrates the best performance with a 0.365 F1 score, improving by 42.0% over SFT and PPO's 0.257.

---

### Note · Authors · 2025-08-11

We sincerely thank reviewers for their positive reviews and initial scores of 5/5/5/4 before rebuttal. We were able to improve the paper further via additional experiments and clarifications following constructive feedbacks by reviewers, and we are looking forward to seeing the updated final scores.

Thank you all for your time and dedication to this process.

---

### Decision · Program_Chairs · 2025-09-17

**Decision:**

Accept (oral)

**Comment:**

This paper introduces QoQ-Med, a multimodal clinical foundation model capable of reasoning over diverse data types, including 2D/3D medical images, ECG signals, and text. The core innovation is Domain-aware Group Relative Policy Optimization (DRPO), a novel reinforcement learning algorithm that addresses data heterogeneity and imbalance in clinical datasets through hierarchical reward scaling. Trained on 2.61 million instruction-tuning pairs spanning nine clinical domains, QoQ-Med significantly outperforms other critic-free training methods such as GRPO in diagnostic performance, particularly in underrepresented domains. Additionally, the model produces interpretable outputs, including reasoning traces and highlighted image regions. The authors have released the full model weights, the modular training pipeline, and all intermediate reasoning traces.

The reviewers unanimously recognize this work as a novel, valuable, and significant contribution to the field. The authors’ rebuttal effectively addressed additional concerns, including statistical significance testing, fairness, and hyperparameter analysis. Given the paper’s originality, importance, and overall quality, I recommend it for an oral presentation to highlight its impactful contributions at the conference.